# Genomic and morphological evidence of distinct populations in the endemic common (weedy) seadragon *Phyllopteryx taeniolatus* (Syngnathidae) along the east coast of Australia

O. Selma Klanten[1]*, Michelle R. Gaither[2], Samuel Greaves[2], Kade Mills[3], Kristine O'Keeffe[4‡], John Turnbull[4,5‡], Rob McKinnon[6‡], David J. Booth[1]

1 Fish Ecology Lab, School of Life Sciences, University of Technology Sydney, Sydney, NSW, Australia, 2 Department of Biology, Genomics and Bioinformatics Cluster, University of Central Florida, Orlando, FL, United States of America, 3 Victorian National Parks Association, Carlton Melbourne, VIC, Australia, 4 Underwater Research Group (URG), Sydney, NSW, Australia, 5 Centre for Marine Biodiversity and Innovation, School of Biological, Earth and Environmental Sciences, University of New South Wales, Sydney, NSW, Australia, 6 New South Wales National Parks and Wildlife Service, Merimbula, NSW, Australia

☯ These authors contributed equally to this work.
‡ These authors also contributed equally to this work.
* osklanten@me.com

**Data Availability Statement:** We note that your Data Availability Statement states that "Fastq

## Abstract

The common or weedy seadragon, *Phyllopteryx taeniolatus*, is an iconic and endemic fish found across temperate reefs of southern Australia. Despite its charismatic nature, few studies have been published, and the extent of population sub-structuring remains poorly resolved. Here we used 7462 single nucleotide polymorphisms (SNPs) to identify the extent of population structure in the weedy seadragon along the temperate southeast coast of Australia. We identified four populations, with strong genetic structure ($F_{ST}$ = 0.562) between them. Both Discriminant Analysis of Principle Components (DAPC) and Bayesian clustering analyses support four distinct genetic clusters (north to south: central New South Wales, southern NSW, Victoria and Tasmania). In addition to these genetic differences, geographical variation in external morphology was recorded, with individuals from New South Wales shaped differently for a few measurements to those from the Mornington Peninsula (Victoria). We posit that these genetic and morphological differences suggest that the Victorian population of *P. taeniolatus* was historically isolated by the Bassian Isthmus during the last glacial maximum and should now be considered at least a distinct population. We also recorded high levels of genetic structure among the other locations. Based on the genomic and to a degree morphological evidence presented in this study, we recommend that the Victorian population be managed separately from the eastern populations (New South Wales and Tasmania).

sequence files were deposited in NCBI's Sequence Read Archive (accession numbers: SRR12395874-12395945) and the associated metadata is available at GEOME and is provided in S1 Table.

**Funding:** Funded study, single grant to OSK and DJB SWR/4/2016 by the Sea World Research and Rescue Foundation Inc. https://seaworld.com.au/conservation The funders had no role in study design, data collection and analysis, decision to publish, or preparation of the manuscript.

**Competing interests:** The authors have declared that no competing interests exist.

## Introduction

Marine fishes often inhabit wide geographic ranges, have large populations with high genetic diversity, and display low levels of genetic structuring due to high larval dispersal capacity [1–5]. Conservation and management of these fish resources is often based on the assumption that population connectivity over large geographic scales confers some level of resilience even in the face of anthropogenic stressors. Under this logic, endemic species with restricted ranges, smaller population sizes, and lower genetic diversity [6] face increased extinction risk due to environmental stressors such as habitat loss and climate change [7, 8]. However, this is not always the case, with some endemic reef fish species displaying high genetic diversity [9–11], theoretically giving them a better chance of adapting to changing environments [12, 13]. Regardless, the conservation of endemic species is important, as their restricted distributions leave them more prone to localised disturbances and the resulting reductions in population size [8, 14–16].

The family Syngnathidae are a unique group of fishes found in temperate and tropical oceans around the world and includes 321 species of seahorses, pipefishes, and seadragons [17, 18]. Their distinct characteristics include a fused elongate jaw, bony plates covering their body like armour, and the absence of pelvic fins [17]. There are only three seadragon species: the highly camouflaged leafy seadragon *Phycodurus eques*, the weedy seadragon *Phyllopteryx taeniolatus* (also known as the common seadragon), and the recently discovered ruby seadragon *Phyllopteryx dewysea* [17, 19–21]. All three are endemic to temperate reefs in southern Australia. The weedy seadragon's status on the IUCN Red List was changed in 2017 from "Near Threatened" (NT) to "Least Concern" (LC) but their population trend is still considered as "Decreasing" (https://www.iucnredlist.org/species/17177/67624517).

The weedy seadragon is a charismatic fish with an unique life history. Like all syngnathids, they exhibit parental care whereby a female deposits her eggs to the male [17]. In the case of weedy seadragons, eggs are deposited under the tail of the male and he carries the developing eggs (30–38 days in the Sydney area until hatching [22]). Before giving live birth, the male moves into shallow sheltered waters, where fully-formed miniature weedy seadragons emerge from the brooded eggs. In contrast to species of broadcast-spawning fish, seadragons do not have a larval dispersal stage. *Phyllopteryx taeniolatus* are weak swimmers and lack the prehensile tail of seahorses, so young seadragons stay close to the substrate, camouflaged and protected by kelp and seaweed. As such, they are less prone than the larvae of broadcast spawning fish to being swept away by currents. Studies on the life history and ecology of weedy seadragons have revealed two major demographic features: firstly, they display high site fidelity [23], only moving 50–500 m as adults and secondly, long-term census data have revealed that their abundance has declined recently at sites near Sydney and Hobart [24]. There are concerns that seadragons are in decline more broadly, possibly due to loss of essential kelp habitat [24–26]. Kelp and other seaweeds, and especially the area at the kelp-sand interface, are important habitat for seadragons, providing shelter in the form of camouflage, as well as supporting their preferred food source, mysid shrimp [17, 27, 28]. A recent study using mtDNA [29] reported strong population structure in *P. taeniolatus* across its range on temperate reefs along the coast of Australia from central New South Wales (NSW) to the southern reefs of Western Australia. This study also reported lower genetic diversity in weedy seadragons from the southeast coast of Australia. The authors [29] suggest that the main barrier between the eastern and western populations of weedy seadragons is the Bassian Isthmus, a land bridge that formed between Tasmania and mainland Australia during the peak of the Last Glacial Maximum (LGM) ca. 19–25 kya and subsequently started to break up to the east ca. 14 kya [29–32]. Another study [33] on the leafy seadragon (*Phycodurus eques*), using both mtDNA and microsatellites, also

found strong population structure especially between western and southern Australian populations, with high to low genetic diversity.

To better resolve the strong genetic structure observed in two mtDNA loci [29] in *P. taeniolatus*, we combined next-generation sequencing with morphological assessments to examine populations along the southeast coast of Australia. In particular, we used a restriction-site associated DNA sequencing (RADseq) approach to generate 7462 single nucleotide polymorphisms (SNPs) from 72 individuals of *P. taeniolatus* from NSW, Victoria, and Tasmania to (1) examine levels of genetic structure and (2) calculate genetic diversity among populations of seadragons, and (3) to compare external morphology among populations using high-resolution images. These aims link to management of this endemic species.

## Materials and methods

University of Technology Ethics committee approved and provided Ethics for weedy seadragon research (field work, study and collection): University of Technology Sydney 2013-553A (2013–2017) and University of Technology Sydney: ACEC ETH 17–1707 (2017–2020).

Sample collection permits (for three states): Department Primary Industries (DPI) New South Wales (NSW) F94/696(A)-8.2 (2012–2019) and additional Jervis Bay Marine Parks permit DPI NSW F94/696(A)-8.3 (2014–2019), Victoria State Dept PA53 (2016–2018), DPI Tasmania 16021 (2016–2017).

### Sample collection

In total 82 *P. taeniolatus* were sampled for genetic analysis between astral summer 2014/15–2018 (S1 Table). A small piece of tissue was collected from one of the leaf-like appendages (Fig 1) from 72 seadragons *in situ* during surveys on temperate reefs along the east coast of Australia from Sydney, New South Wales (NSW) to the Mornington Peninsula in Victoria (VIC) and the east coast of Tasmania (TAS; Fig 1). Another 10 tissue samples (appendages; from 2015–2018), were taken from dead individuals that had washed up onshore. During some of our ongoing *in situ* surveys (2018–2019), we also took high-resolution images of individual weedy seadragons with a measuring tape or ruler placed directly behind or above an individual for morphological comparisons. In total we measured 45 individuals, 20 each from NSW and VIC and 5 from TAS (S2 Table).

### Genotyping

**RADSeq library preparation and sequencing.** Genomic DNA was extracted at the Sydney Institute of Marine Science (SIMS) from each of the 82 samples using the Isolate II Genomic DNA Kit (Bioline) following the manufacturer's protocol. DNA was quantified using a nanodrop and sent to the Australian Genome Research Facility (AGRF) for library preparation and sequencing. RADSeq libraries were prepared following the double digest RADSeq protocol of Peterson et al. [34] using the enzymes pstI and hpyCH4IV. Following digestion of genomic DNA, barcoded adapters were ligated onto fragments. Libraries were then pooled and all fragments between 280 and 342 bp were selected using a Blue Pippin (Sage Science). Libraries were then amplified via PCR with indexed primers and sequenced on an Illumina NextSeq 500 using 1x150 cycles in mid-output mode. The RADSeq data generated by AGRF used single barcodes that are incorporated into the names of the Fastq files for each individual. Fastq sequence files were deposited in NCBI's Sequence Read Archive (accession numbers: SRR12395874-12395945) and the associated metadata is available at GEOME [35] and is provided in S1 Table. Data generated as vcf file (S1 Data) and as genepop file (S2 Data).

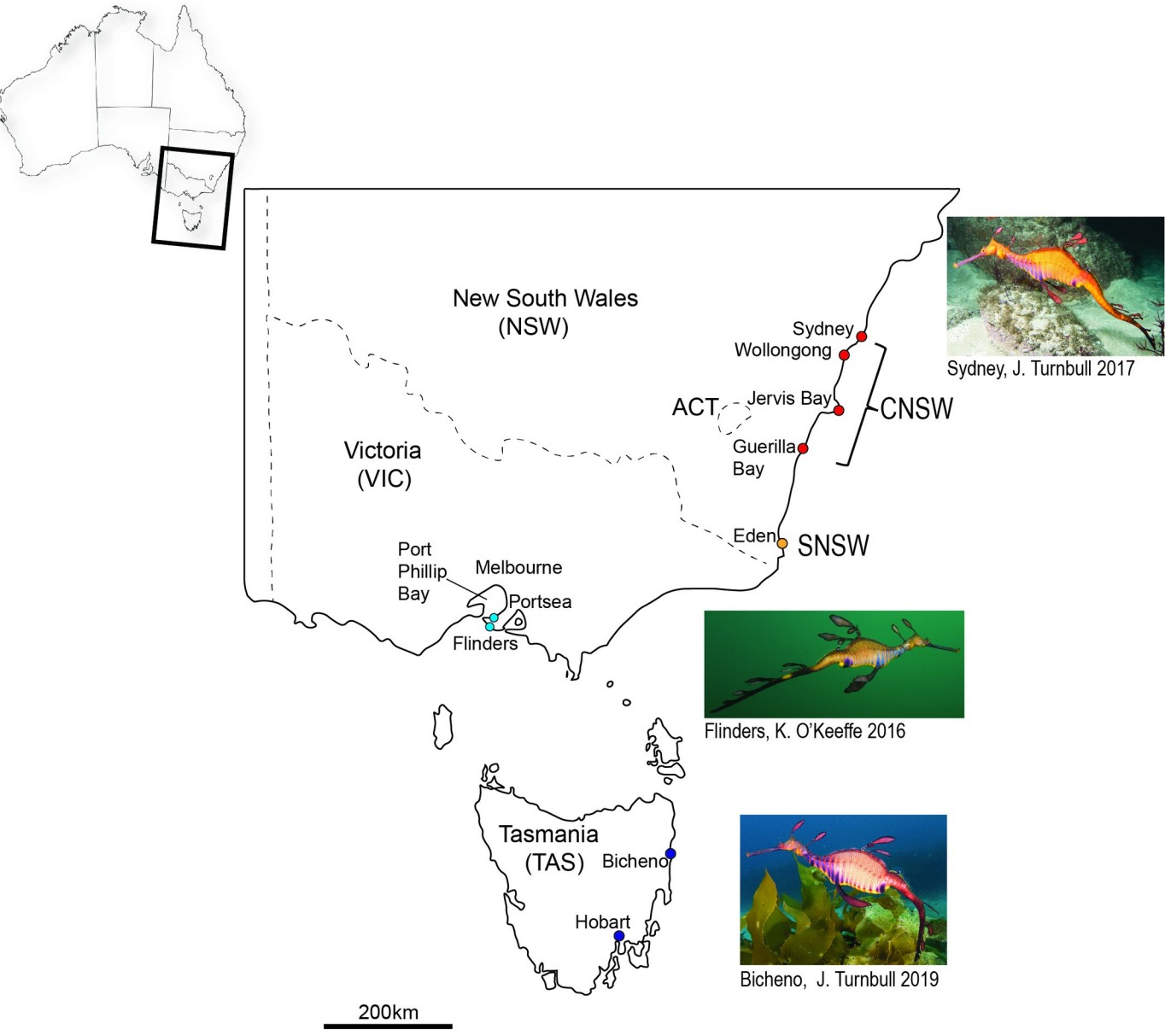

**Fig 1. Map of sample sites.** CNSW (Central New South Wales, red dots) with Sydney N = 13 and Botany Bay N = 19, Wollongong N = 1, Jervis Bay N = 4, Guerilla Bay N = 3; SNSW (Southern New South Wales, yellow dot) with Eden N = 8; VIC (Victoria, Mornington Peninsula, light blue dots) with Flinders Pier N = 10, Portsea N = 5; and TAS (Tasmania, dark blue dots) with Bicheno N = 7 and Hobart N = 2. Photos are of adult specimens of the weedy seadragon, *Phyllopteryx taeniolatus* from three of the sample sites (with photo credit given).

**Data analysis.** Demultiplexing of the sequence data and initial quality control was conducted at AGRF with STACKS v.1.41 [36, 37] using default settings. Only sequences with exact barcodes were retained and sequences were trimmed to 140 bp. All other data processing took place at the Genomics and Bioinformatics Cluster at the University of Central Florida using STACKS v.2.0b. Loci were assembled using the denovo pipeline using default settings except we allowed for four mismatches between stacks within individuals (-*M*) and four mismatches between sample loci when building the catalog (-n). See Rochette et al. [38] for a detailed explanation of the Bayesian genotype caller employed in STACKS. Stacks in which the

numbers of reads were more than two standard deviations above the mean were assumed to be repetitive elements and removed from the data set. To eliminate poor quality samples, those individuals with fewer than 200,000 reads or less than 60% of loci were excluded from further analysis, resulting in a final sample size of 72 individuals. Using the '*populations*' function in STACKS, we eliminated those loci with observed heterozygosity greater than 0.7 (—*max-obs-het*) and applied a minor allele frequency cut-off of 0.05 (—*min-maf*). We retained only those loci that amplified in $\geq 80\%$ of individuals (-r). Output files were created by implementing the '—*write_single_snp*' option which outputs the first SNP per locus.

Genetic diversity (observed heterozygosity $H_o$, expected heterozygosity $H_e$, inbreeding coefficient $F_{IS}$, and allelic richness $A_R$) was calculated for the overall sample set, sample locations, and later the putative populations using the R package pegas [39] and Genepop [40]. Due to small samples size Wollongong (N = 1) and Hobart (N = 2) were omitted from population level calculations. To test for hierarchical population structure, we ran an Analysis of Molecular Variance (AMOVA) using the program Arlequin v.3.5 [41] with significance determined using 50,000 permutations. Pairwise $F_{ST}$ values among the populations with N > 5 individuals were calculated in Arlequin v.3.5, excluding Wollongong and Hobart. We corrected for multiple comparisons according to Benjamini and Yekutieli [42] by controlling false discovery rates using the *p.adjust* function in R. In addition, we tested for Isolation by Distance (IBD) [43] using a Mantel test and 100,000 permutations, as implemented in the R package ade4 v.1.7–15 [44]. For this analysis we used the pairwise $F_{ST}$ calculated in Arlequin and geographic distances using Google Earth following the Eastern Australian Current (EAC) along the east coast to Tasmania using the Ruler tool Path by hand. In addition, two distances were calculated to the Mornington Peninsula (VIC) population, one as an open (present day) connection, second following a closed land bridge (Bassian Isthmus) around TAS towards VIC.

To explore the possibility of population structuring, we used two methods, Bayesian clustering algorithms of STRUCTURE and Discriminant Analysis of Principle Components (DAPC). First we ran STRUCTURE v.2.3.4 [45]. We tested *K* 1–6, with nine independent runs for each *K*, using a burn-in of 100,000 iterations, followed by 500,000 Markov chain Monte Carlo (MCMC) iterations. The results were combined and the most likely *K* was determined using the Evanno method [46] in STRUCTURE HARVESTER [47] (S1A Fig) for result of *K*). Data were visualized using the Cluster Markov Packager Across *Ks* (CLUMPAK) online tool [48]. Secondly, we conducted a Discriminant Analysis of Principle Components (DAPC) using the R package ADEGENET v.2.2.1 [49]. Because PCA analyses require no missing data (and DAPC uses PCA) we replaced missing data with the mean allele frequency for each locus using the function *scaleGen* in ADEGENET. Clusters were generated using the *find.clusters* function, which uses the k-means algorithm to generate clusters for various values of k, and then Bayesian Information Criterion (BIC) to determine the most likely value of k (S1B Fig). Following the results of the cross-validation algorithm using 500 iterations, we retained 12 principal components and three discriminant functions for the DAPC. For comparison we also ran Principle Components Analysis (PCA) using the R package ade4 [44] using the DAPC data file.

After putative populations were defined, we tested for Hardy-Weinberg equilibrium (HWE) and linkage disequilibrium (LD) using the R packages pegas [39] and genetics (https://cran.r-project.org/package=genetics), respectively. Loci were excluded if they were significantly out of HWE or in linkage disequilibrium in at least two of the four putative populations. After controlling false discovery rate [42], 81 loci (1 out of HWE and 80 in LD) were removed from further analysis.

## Morphological measurements

Forty-five adult *P. taeniolatus* individuals (32 *in situ*, 13 dead) were measured using ImageJ v.152a [50] from high-resolution images. We modified Forsgren and Lowe [51] measurements of a seahorse, to examine morphological differences between weedy seadragons of New South Wales, Victoria and Tasmania (Fig 2). We explored location differences in body shape using Principal Component Analysis (PCA) using a variance -covariance transformation matrix (to scale all metrics) and Linear Discriminant Analysis (LDA) to investigate natural group assignments with PAST v.4.03 [52]. In case any differences were simply allometric patterns, we plotted each morphological attribute against total length, separately for NSW, TAS, and VIC. Where possible the sex of specimen (29 individuals of 45 specimens measured) was added to our table (S2 Table).

## Results

A total of 19608 loci were resolved across our final set of 72 individuals, of which 7462 were variable and passed filtering. Samples had an average coverage of 21 reads per SNP with a data matrix that was 92.6% complete (see S1 Table for % missing loci per individual). The overall AMOVA based on the nine seadragon populations examined here detected high levels of population structure ($F_{ST} = 0.561$, $p < 0.0001$; S3A Table) along the southeast coast of Australia. When grouping all nine populations into four areas, central New South Wales (CNSW; Sydney, Botany Bay, Wollongong and Jervis/Guerilla Bays), southern NSW (SNSW; Eden),

### Adult *Phyllopteryx taeniolatus*

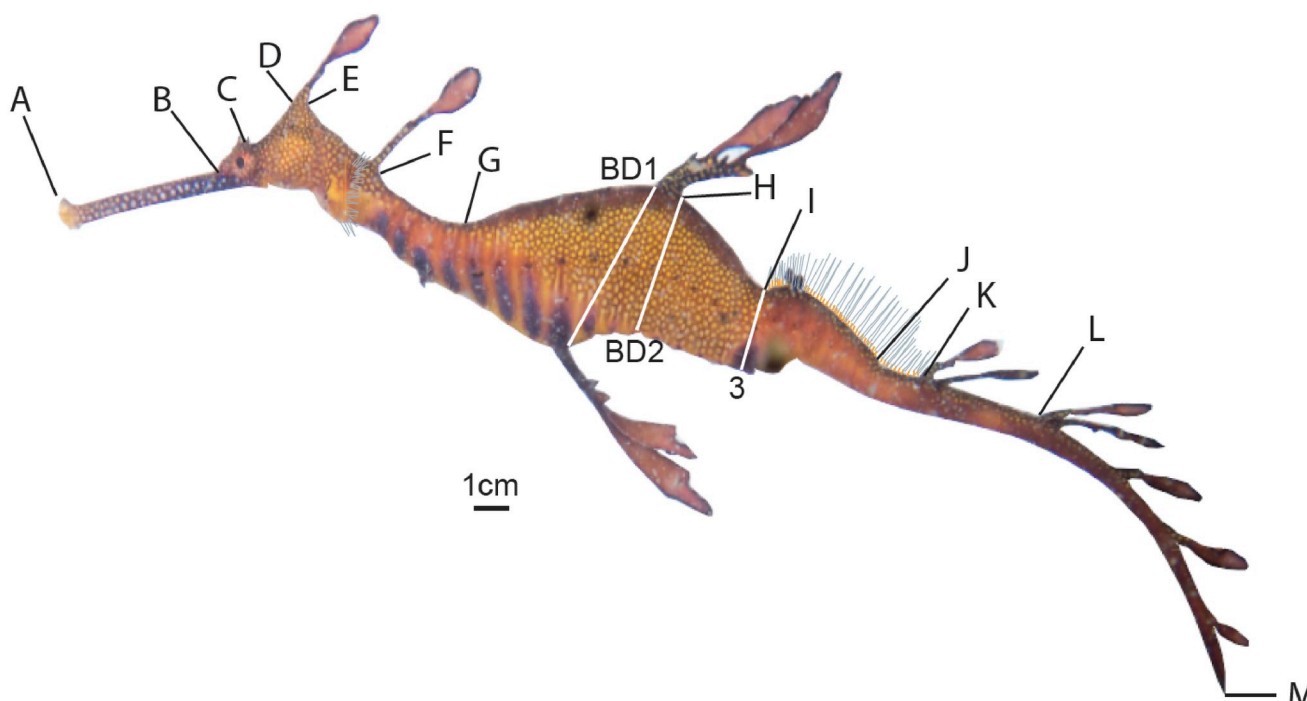

**Fig 2. Morphological measurements taken for adult weedy seadragons *Phyllopteryx taeniolatus*.** Measurements were modified from Forsgren & Lowe [51]. Point to point distance measures are: A-B: snout tip-snout end; B-C: snout end-eye ridge; C-D: eye ridge-base head; D-E: base head-posterior head; E-F: posterior head-base nape; F-G: base nape-nape, G-H: nape-base dorsal; H-I: base dorsal-anterior fin; I-J: anterior fin-posterior fin; J-K: posterior fin-base 1 tail; K-L: base 1 tail-base 2 tail; L-M: base 2 tail-end of tail; BD1: Body depth; BD2: Body depth; 3: Split between body and tail.

**Table 1. Pairwise $F_{ST}$ values between populations of *P. taeniolatus*.**

| | CNSW | | | SNSW | TAS | VIC |
|---|---|---|---|---|---|---|
| | Sydney | Botany Bay | Jervis/ Guerilla | Eden | Bicheno | Flinders/Portsea |
| Sydney | - | 1.000 | 0.002 | 0.000 | 0.003 | 0.000 |
| Botany Bay | 0.002 | - | 0.000 | 0.000 | 0.000 | 0.000 |
| Jervis/Guerilla | **0.155** | **0.169** | - | 0.021 | 0.026 | 0.000 |
| Eden | **0.525** | **0.524** | **0.392** | - | 0.015 | 0.000 |
| Bicheno | **0.441** | **0.443** | **0.334** | **0.494** | - | 0.000 |
| Flinders/Portsea | **0.690** | **0.703** | **0.636** | **0.592** | **0.649** | - |

$F_{ST}$ values are of populations with > 5 individuals (excluding Wollongong in NSW and Hobart in TAS). Below the diagonal are $F_{ST}$ values calculated using Arlequin v.3.5 [41] while above the diagonal are *p*-values corrected for false discovery rates [42]. Significant values are bolded. CNSW–central NSW, SNSW–southern NSW, TAS–Tasmania and VIC–Victoria (combined both sites).

Victoria (VIC; Flinders Bay, Portsea) and Tasmania (TAS; Bicheno and Hobart), the AMOVA result was $F_{ST}$ = 0.630 (*p*<0.0001; S3B Table).

All pairwise $F_{ST}$ values among seadragon populations were highly significant and ranged from 0.155 to 0.703 with the exception of populations from Sydney and Botany Bay, which were not significantly different (Table 1). Each population harboured private alleles including CNSW (1721), SNSW (348), TAS (563), and VIC (3918). While none of these were fixed, a substantial proportion reached an allele frequency of 0.9 meaning they were nearly fixed (CNSW: 1200, 70%; SNSW: 128, 37%; TAS: 161, 29%; VIC: 1385, 35%).

Among the populations in NSW, the southern population of Eden (SNSW) showed significant structure when compared to Sydney, Botany Bay, Jervis/Guerilla Bays (CNSW) with pairwise $F_{ST}$ ranging from 0.392 to 0.525 (Table 1). The Tasmanian population of Bicheno was also distinct from Flinders Bay and Portsea in Victoria ($F_{ST}$ = 0.649) and populations in NSW ($F_{ST}$ ranging from 0.334–0.494). Victorian samples from Flinders/Portsea were strongly structured compared to populations in NSW and Bicheno population in TAS ($F_{ST}$ ranging 0.636–0.703; Table 1).

Mantel tests indicated a significant correlation of $F_{ST}$ for an open connection (r = 0.654, *p* = 0.026) and closed land bridge connection (r = 0.820, *p* = 0.014) with geographic distance, suggesting that isolation by distance was likely responsible for the observed population segregation (S2 Fig).

Furthermore, genetic diversity based on SNPs resulted in an overall observed heterozygosity of $H_o$ = 0.220, expected heterozygosity $H_e$ = 0.302, allelic richness $A_R$ = 1.907 and an inbreeding coefficient of $F_{IS}$ = 0.113 (Table 2). Seadragons from CNSW and TAS had lower expected heterozygosity ($H_e$ = 0.136 and $H_e$ = 0.140, respectively) compared to the VIC population ($H_e$ = 0.209). In contrast, inbreeding coefficient was lower in VIC ($F_{IS}$ = 0.079) compared to CNSW ($F_{IS}$ = 0.112) and TAS ($F_{IS}$ = 0.143). Allelic richness ($A_R$) was similar for all populations (Table 2).

Results from STRUCTURE HARVESTER (S1 Fig) indicated three genetic clusters, while DAPC analyses indicated four (Figs 3 and 4, respectively). However, individual assignment for K = 4 STRUCTURE runs aligned with those generated by DAPC. The finding of multiple clusters was also supported by the AMOVA (S3 Table) and PCA analyses (S3 Fig).

Interestingly, based on STRUCTURE, all (N = 7) individuals from Jervis/Guerilla Bays (CNSW) appeared to have some shared genetic loci with Eden (SNSW) and Tasmania (Fig 3, K = 4). Additionally, two individuals from Hobart (TAS) shared loci with CNSW.

Several differences in external morphology between seadragons from VIC, TAS and NSW were clear from the biplots (S4 Fig) and populations were resolved using LDA (Fig 5A) but not

**Table 2. Genetic diversity indices for *P. taeniolatus*.**

| | Population | N | $H_o$ | $H_e$ | $F_{IS}$ | $A_R$ |
|---|---|---|---|---|---|---|
| CNSW | Sydney | 14 | 0.119 | 0.120 | 0.027 | 1.374 |
| | Botany Bay | 18 | 0.119 | 0.120 | 0.016 | 1.377 |
| | Wollongong | 1 | - | - | - | - |
| | Jervis/Guerilla | 7 | 0.152 | 0.165 | 0.174 | 1.452 |
| | All | 40 | 0.129 | 0.136 | 0.112 | 1.521 |
| SNSW | Eden | 8 | 0.154 | 0.155 | 0.024 | 1.425 |
| TAS | Bicheno | 7 | 0.129 | 0.130 | 0.025 | 1.332 |
| | Hobart | 2 | - | - | - | - |
| | All | 9 | 0.131 | 0.140 | 0.143 | 1.377 |
| VIC | Flinders/Portsea | 15 | 0.201 | 0.209 | 0.079 | 1.575 |
| Overall | | 72 | 0.220 | 0.302 | 0.113 | 1.907 |

Diversity indices are based on all 19608 loci, along the southeast coast of Australia; sample size (N), observed heterozygosity ($H_o$), expected heterozygosity ($H_e$), inbreeding coefficient ($F_{IS}$) and allelic richness ($A_R$). Due to low sample size Wollongong (from CNSW) and Hobart (TAS) were omitted from analyses.
Population description: CNSW central New South Wales, SNSW southern NSW, TAS Tasmania, VIC Victoria.

as clear from PCA (Fig 5B). The PCA pattern was somewhat driven by body length along PC1 (Fig 5B eigenvalues), while LDA had a 96% correct classification for all individuals into 3 groups (NSW, TAS, VIC). Seven out of 15 body measurements scaled by total length distinguished morphological attributes between NSW and VIC. The shapes along the head (Fig 2D–2F), neck (Fig 2F and 2G), the main body (Fig 2H–2J) and body depths (Fig 2BD1 and 2BD2) differed among populations according to the biplots (S4 Fig). Most NSW individuals (16 specimens out of total 20) were not sexed and VIC specimens were mostly males with only four females (out of total 20). Overall, only 10 females were measured out of total 45 specimens (S2 Table). The few females were indicated separately by location in our biplots for body depth (BD1 and BD2) and showed minimal influence (S4 Fig).

## Discussion

The endemic *P. taeniolatus* population is highly structured along the southeast coast of Australia based on a genome-wide set of SNPs. In fact, four genetically distinct populations of weedy seadragon populations are recognised here: central New South Wales (CNSW), southern New South Wales (SNSW), Victoria (VIC) and Tasmania (TAS) across 10 degrees of latitude. Our results report a high degree of genetic separation similar to that found in a study based on just two mtDNA loci [29], which also reported some shared haplotypes of seadragon populations between NSW and Tasmania. High site fidelity combined with lack of a dispersal phase of juveniles and overall weak swimming ability most likely resulted in the highly genetically-structured populations of *P. taeniolatus*, which is further supported by isolation by distance (IBD) results. Of note is that all individuals from two sites in Victoria (Flinders Bay and Portsea) were highly distinct, genetically and to a degree morphologically, to the main populations of central and southern NSW, as well as Tasmania [see also 29].

The Bassian Isthmus acted as a barrier to marine dispersal for 20–25 kya during the last glacial maximum (LGM) [29–31], and may have prevented mixing between these populations, providing the conditions for Victorian weedy seadragons to diverge in isolation. Since the opening of the Bassian Isthmus, approximately 14 kya, there has been ample opportunity for gene flow, therefore the question arises of whether the Victorian seadragon populations are reproductively isolated. While individuals from Victoria were somewhat smaller, biplots (S4

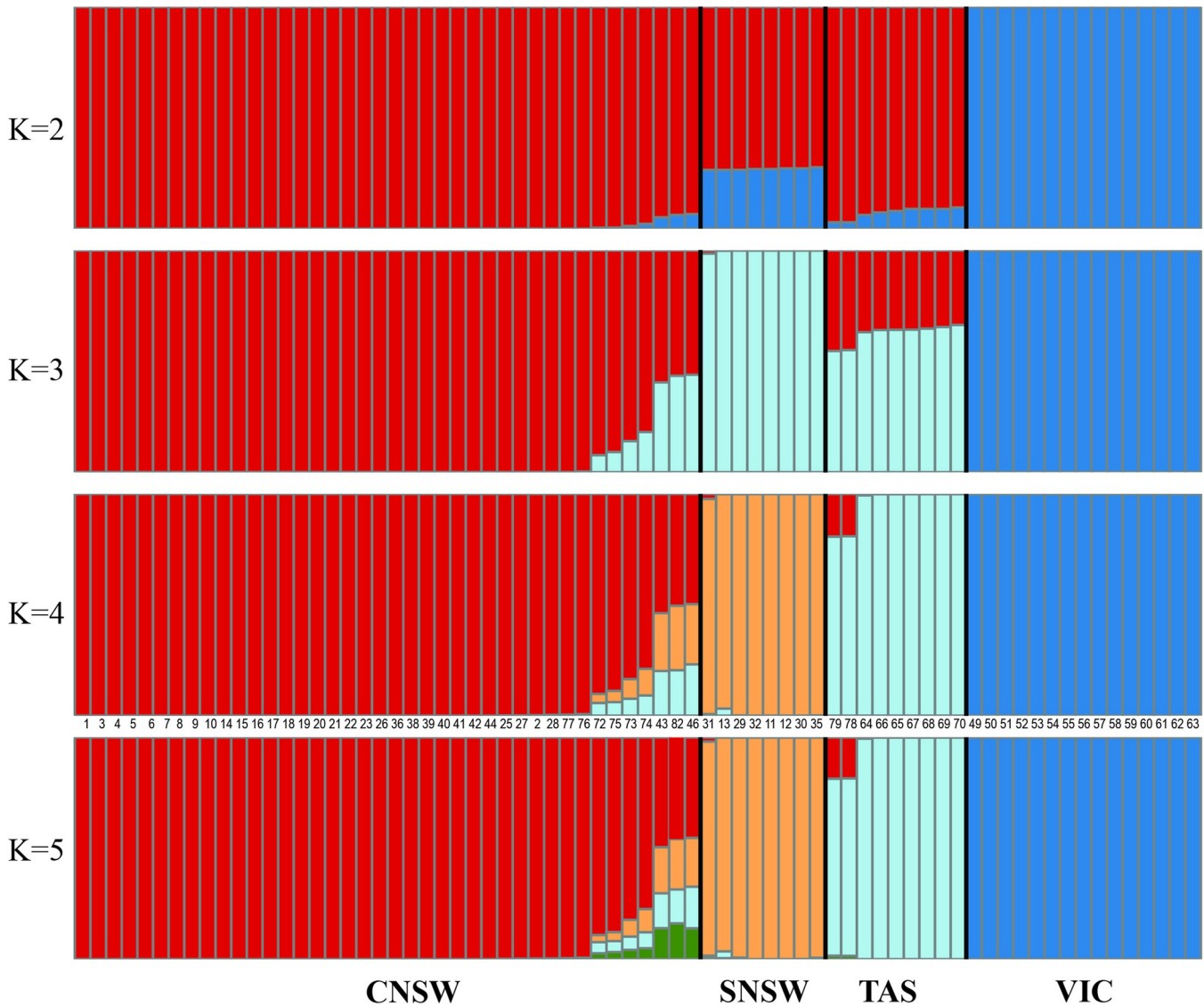

**Fig 3. STRUCTURE results for *P. taeniolatus* from southeast Australia.** Result from the Bayesian clustering algorithms of STRUCTURE v.2.3.4 [45]. Results for *K* 2–5 are shown. Individuals from Wollongong (N = 1) and Hobart (N = 2) were included in these analyses. The most likely *K* was four and represent the four distinct populations: central New South Wales (CNSW), southern New South Wales (SNSW), Tasmania (TAS), and Victoria (VIC). Numbers below *K* = 4 are individual seadragon sample IDs (see S1 Table, under Field ID).

Fig) indicate that shape differences summarised in Fig 5A were not simply length-related. Unfortunately, we were unable to measure differences in sexual dimorphism because few of our specimens were female, but such data will be valuable to gather in the future.

Considering these two lines of evidence, genome-wide SNP data and morphology, we suggest the Victorian *P. taeniolatus* to be a highly distinct population that is unlikely to be currently interbreeding with the eastern populations (NSW, TAS) and must be managed separately [53]. Our isolation by distance results potentially show the genetic impact of this division (S2 Fig). Isolation over similar distances was found in the related species *Hippocampus erectus* (the lined seahorse) from the western Atlantic Ocean. SNP analysis showed three highly distinct populations (Gulf and Keys, Florida Atlantic coast, northern Atlantic coast) and suggested a persistent isolation of the northern populations [54].

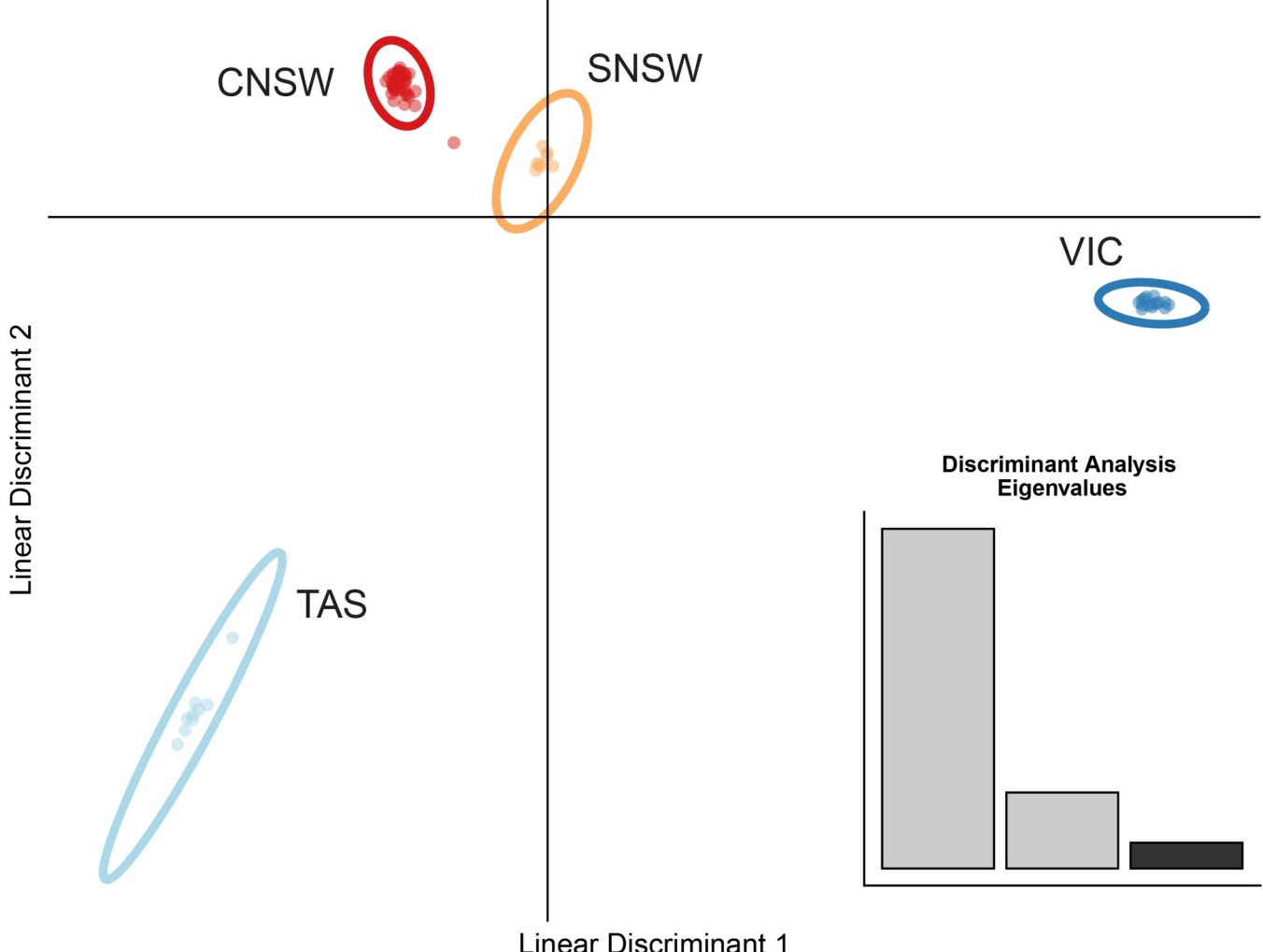

**Fig 4. DAPC result for *P. taeniolatus*. Discriminant analysis of principal components (DAPC) calculated using the R package ADEGENET v.2.2.1 [49].** Individuals from Wollongong (N = 1) and Hobart (N = 2) were included in these analyses. The results support four separate genetic clusters representing central New South Wales (CNSW), southern New South Wales (SNSW), Victoria (VIC) and Tasmania (TAS).

We suggest that due to the degree of genetic distinctiveness [53, 55] and its association with some morphological differences, the weedy seadragons along the temperate coast of Australia should be evaluated for the presence of a subspecies [53, 55–57]. This in turn, will have considerable management and conservation issues, with Victorian seadragons the main representatives of this species in the global aquarium trade. Our results show similarly high levels of population structure to a previous study which sampled across the entire species distribution, but relied only on two mitochondrial genes [29]. The study focused on a possible genetic break across the Great Australian Bight, but did also show divergence on either side of the Bass Strait [29]. The thousands of genomic markers used here are a robust sampling of the nuclear genome, and reflect the evolutionary history of these populations with greater accuracy than the maternally inherited mitochondrial genome taken alone [53]. Together, these results show substantial genetic divergence across the Bass Strait at levels considered evolutionarily significant in other taxa [53, 55, 56]. A comparative genome study on 61 pairs of populations/species of animals [55] showed splits between populations and species with comparable pairwise $F_{ST}$ values to this study (on average 0.665 with many values lower).

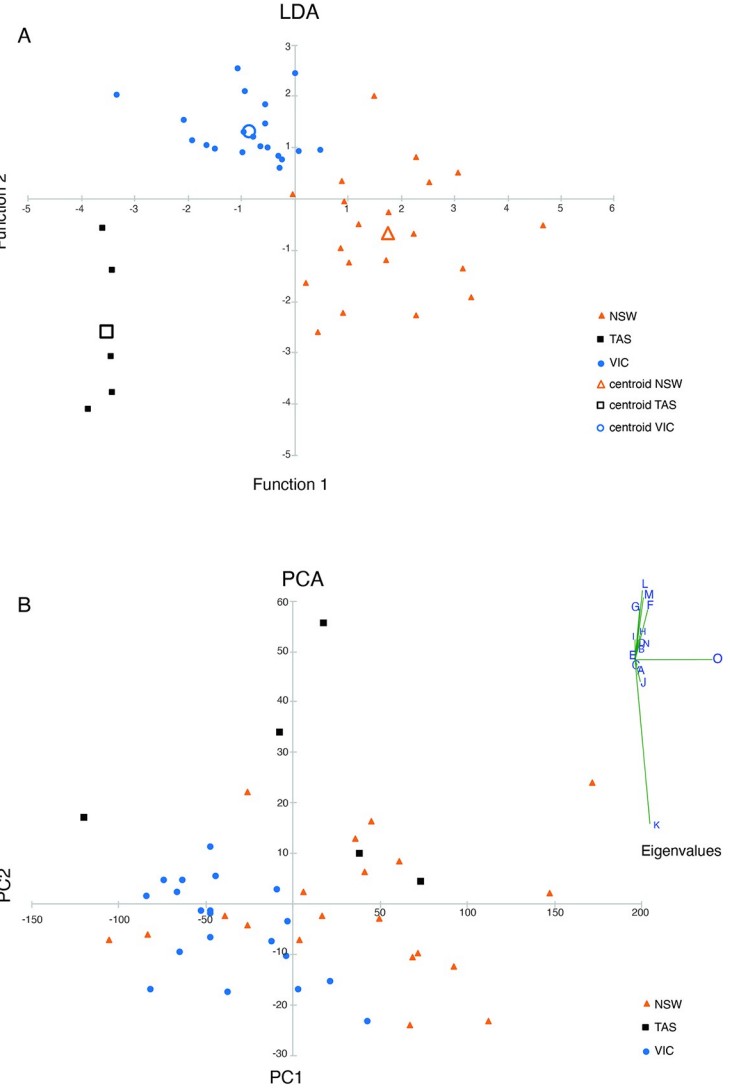

**Fig 5. Linear Discriminant Analysis (LDA) and Principal Component Analysis (PCA) of morphological measurements.** (A) LDA biplot of Function 1 vs Function 2 with centroids indicated, and (B) PCA biplot of PC1 vs PC2, loadings of each seadragon specimen using the suite of external metrics from Fig 2. Eigenvalues are indicated for PCA, prominent eigenvalues are total length (O), measure of tail (K) and two body depth (L, M). Symbols: Brown triangles NSW, blue circles VIC, and black squares TAS specimens. Centroids of LDA are indicated for each group are open using same shape and colour.

Seadragons from Tasmania are similar to NSW seadragons in morphology and habitat use. Weedy seadragons from NSW and TAS are usually found between 10–20 m on rocky reefs with kelp compared to seadragons from the Mornington Peninsula (VIC), which are usually found shallower between 3–10 m depth on soft bottom with seagrass [17]. Furthermore, there is some evidence of potential interbreeding among individuals sampled from Jervis/Guerilla Bays (CNSW), with the Eden (SNSW) and TAS populations. It is possible that over an evolutionary time frame, interbreeding has been facilitated by the East Australian Current (EAC) which could transport seadragons rarely washed offshore between these distant populations. In contrast, individuals of *P. taeniolatus* from both locations in Victoria (Flinders Bay and Portsea) show no signs of admixture with any other individuals sampled across NSW and Tasmania (see also [29]). There are limited signatures of shared ancestry between the Victorian

and eastern populations (Fig 3), but this likely is due to unidirectional, infrequent gene flow eastward through the Bass Strait along the Victorian coast.

Comparing genetic diversity of the endemic *P. taeniolatus* to related species of syngnathids is difficult because there are few studies in these taxa using SNP data. However, based on two mitochondrial genes, genetic diversity of the weedy seadragon on the southeast coast of Australia has been reported to be low [29]. In addition, the leafy seadragon, *Phycodurus eques* of western Australia also reported low genetic diversity (overall $H_e$ = 0.354) based on seven microsatellites [33]. This low genetic diversity in both the endemic weedy (this study) and leafy [33] seadragons stands in comparison with other members of the family Sygnathidae, which have moderate to high heterozygosity. Specifically, $H_e$ ranged from 0.452–0.949 in five studies of widespread seahorses (*Hippocampus* sp.; [58–61]) and from 0.752–0.958 in four widespread pipefishes (*Syngnathus* sp.; [62–65]). These microsatellite data should be interpreted with caution, as their mutation rates and maximum values are different. Theoretically, the maximum heterozygosity derived from biallelic loci (such as the SNPs employed in this study) is 0.5 while it can approach 1.0 with many alleles in a microsatellite locus.

Other studies on endemic marine fish species over similar distribution ranges as this study and based on SNPs, also reported similar low levels of genetic diversity [6, 66, 67]. They suggest that endemic species with small population sizes do display lower genetic diversity compared to widespread species. This does not imply that endemic species are less resilient under normal circumstances, but instead have a smaller gene pool due to smaller population sizes and restricted distribution ranges [9, 13]. If, as our study suggests, the endemic *P. taeniolatus* is formed of highly differentiated populations with overall low genetic diversity and little to no interbreeding, there are implications for management and conservation, as this species is currently considered "Least Concern" but "Decreasing" on the IUCN Red List (2017). The current prognosis is that due to an increase in anthropogenic stressors resulting in habitat loss (kelp and seagrass) and an increase of sea surface temperature [25, 68, 69] the decrease in abundance may lead to further loss of genetic diversity in future generations of weedy seadragons.

## Conclusion

Combining genomics and morphological data, we demonstrate that weedy seadragon populations are highly structured along the southeast coast of Australia. We therefore recommend that the four distinct populations, especially the Victorian population of the Mornington Peninsula, should be managed separately with a precautionary approach and accordingly by each state (VIC, NSW and TAS). Furthermore, we suggest that seadragons from Victoria, isolated and divergent from the eastern populations, warrants a full evaluation as to whether it represents a subspecies of *P. taeniolatus*. In addition, as VIC populations are the main source used for the aquarium trade and in breeding programs, a bias towards this population exists in captivity. To better understand the distinctiveness of the VIC population, behavioural experiments should be undertaken to clarify if it is reproductively isolated, and also if interbreeding seadragons in southern NSW exhibit lower fitness. It appears that females are highly selective in mate choice, i.e. with whom to deposit their eggs, and successful breeding of weedy seadragons is rare and difficult (pers. comm Melbourne Aquarium). Low levels of genetic diversity with highly restricted gene flow indicate that the endemic *P. taeniolatus* may lack resilience in face of future anthropogenic stressors, thus careful population assignment and management is critical.

## Supporting information

**S1 Fig.** (A) Evanno delta-K plot. Results from STRUCTURE HARVESTER run on 72 individuals of the seadragon *P. taeniolatus* from southeastern Australia. (B) Bayesian Information

Criterion (BIC) plot. Results from *find.clusters()* method run in ADEGENET on 72 individuals of the seadragon from southeastern Australia. BIC value decreases until k = 4, then begins to slightly increase at k = 5, indicating 4 is the most likely number of clusters.
(TIF)

**S2 Fig. Isolation by distance (IBD).** IBD results for (a) open connection (no land bridge) and (b) closed connection (Bassian Isthmus historical land bridge). Blue triangles are pairwise comparisons including Victorian populations, grey circles are all other pairs.
(TIF)

**S3 Fig. Principle component analyses (PCA).** (A) Results of PC1 and PC2 and (B) PC2 and PC3 were run on 72 individuals of seadragon from southeastern Australia using the R package ade4.
(TIF)

**S4 Fig. Biplots of morphological data.** All 15 measurements were scaled by total length for each specimen (N = 45). Individuals were grouped into three groups representing 3 states NSW, VIC and TAS. Trendlines were added to each group. Note that TAS had only N = 5 samples. For body depth (BD1 and BD2) we indicate where possible sex, however dead individuals could not be sexed (majority of NSW specimens N = 13, and 3 from Eden). Symbols: Brown triangle NSW (open brown triangle female N = 2 and open brown rhomboid male N = 2), blue circle VIC male (open blue circle female N = 4) and black square TAS male (open black square female N = 4).
(ZIP)

**S1 Table. Complete metadata for 72 *P. taeniolatus*.** Data for seadragons from southeastern Australia from which genetic data was generated. It includes % missing loci, number of reads and average coverage.
(XLSX)

**S2 Table. Morphology measurements.** Data for all 15 body measurements of adult *P. taeniolatus* see Fig 2 for details. Where possible male and female of an individual seadragon is indicated.
(XLSX)

**S3 Table. AMOVA results.** AMOVA of (A) all nine *P. taeniolatus* populations, and (B) grouped into four populations (CNSW, SNSW, VIC and TAS).
(DOCX)

**S1 Data. Data generated as vcf file.**
(ZIP)

**S2 Data. Data generated as genepop file.**
(GENEPOP)

## Acknowledgments

We would like to thank citizen scientists: Tim Green, Noah and Brook Love, Casey Gibson and Donna Powel who provided us with washed-up dead seadragons used in this study. We thank the staff at the molecular laboratories at Sydney Institute of Marine Science (SIMS) for their support. We would also like to thank two anonymous reviewers for their constructive comments on this manuscript.

## Author Contributions

**Conceptualization:** O. Selma Klanten.

**Data curation:** O. Selma Klanten, Michelle R. Gaither.

**Formal analysis:** O. Selma Klanten, Michelle R. Gaither, Samuel Greaves, Kade Mills, David J. Booth.

**Funding acquisition:** O. Selma Klanten, David J. Booth.

**Investigation:** O. Selma Klanten.

**Project administration:** O. Selma Klanten.

**Resources:** O. Selma Klanten, Michelle R. Gaither, Samuel Greaves, Kade Mills, Kristine O'Keeffe, John Turnbull, Rob McKinnon, David J. Booth.

**Supervision:** O. Selma Klanten, Michelle R. Gaither, David J. Booth.

**Validation:** O. Selma Klanten, Michelle R. Gaither, Samuel Greaves.

**Visualization:** O. Selma Klanten, Michelle R. Gaither.

**Writing – original draft:** O. Selma Klanten, Michelle R. Gaither, Samuel Greaves, Kade Mills, John Turnbull, David J. Booth.

**Writing – review & editing:** O. Selma Klanten, Michelle R. Gaither, Samuel Greaves, Kade Mills, Kristine O'Keeffe, John Turnbull, Rob McKinnon, David J. Booth.

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
