## [Decision Letter · Decision Letter 0]

17 Jul 2020

PONE-D-20-17523

Genomic and morphological evidence of distinct subpopulations in the endemic weedy seadragon Phyllopteryx taeniolatus (Syngnathidae) along the east coast of Australia

PLOS ONE

Dear Dr. Klanten,

Thank you for submitting your manuscript to PLOS ONE. After careful consideration, we feel that it has merit but does not fully meet PLOS ONE’s publication criteria as it currently stands. Therefore, we invite you to submit a revised version of the manuscript that addresses the points raised during the review process.

We look forward to receiving your revised manuscript.

Kind regards,

Tzen-Yuh Chiang

Academic Editor

PLOS ONE

Journal Requirements:

3. Please include table 3 as part of your main manuscript and remove the individual file.

Please note that supplementary tables should remain as separate "supporting information" files.

Reviewers' comments:

Reviewer's Responses to Questions

**Comments to the Author**

1. Is the manuscript technically sound, and do the data support the conclusions?

Reviewer #1: Yes

Reviewer #2: Partly

2. Has the statistical analysis been performed appropriately and rigorously? 

Reviewer #1: Yes

Reviewer #2: Yes

3. Have the authors made all data underlying the findings in their manuscript fully available?

Reviewer #1: Yes

Reviewer #2: No

4. Is the manuscript presented in an intelligible fashion and written in standard English?

Reviewer #1: Yes

Reviewer #2: Yes

5. Review Comments to the Author

Reviewer #1: Klanten et al. “Genomic and morphological evidence of distinct subpopulations in the endemic weedy seadragon Phyllopteryx taeniolatus (Syngnathidae) along the east coast of Australia”.

Summary

Klanten et al., present a population genetics study on the weedy seadragon (Phyllopteryx taeniolatus) using SNP sequence data generated from ddRADseq. This work builds on previous studies that used mitochondrial and microsatellite sequence data. In this study the authors have contributed further evidence for significant population structuring at a regional scale that has implications for the management of this species. They have also incorporated morphological measurements as further evidence of population isolation and interpret these results along with the genetic data with reference to the historical Bassian Isthmus as a barrier to dispersal.

The SNP data generated in this study have been assembled and filtered appropriately to produce a high-quality dataset for which to analyse genetic diversity and patterns among the populations. The authors have used multiple different analyses to test for genetic population structure, and these have been carried out appropriately for the dataset. The phylogenetic analyses are not suitable for this data set however, and should be removed (comments below). There is also widespread unjustifiable use of the word hybridisation, which needs to be removed. Otherwise, this is a robust contribution to understanding these iconic fish.

1. Line 59: “…(taxonomically known as the common seadragon)” isn’t correct as common names are not controlled by the ICZN. Suggest change to: (also known as the common…).

2. Line 62: Ruby seadragons are thought to have prehensile tails (Rouse et al. 2017 First live records of….Mar Bio Records).

3. Lines 77-88: You state that mtDNA showed strong population structure for weedy seadragons, and lowered genetic diversity on the east coast. Yet at the end of this paragraph you state that leafy seadragon genetic diversity was moderate to low, a similar pattern to the weedy. This last statement is in contrast with [28] which actually reports some areas with quite high diversity, but showed low diversity for the east coast. You either need to elaborate further on this complexity, or just shorten the final sentence of the paragraph so it does not include a species comparison.

4. Line 149: Were pairwise Fst values also calculated in Arlequin or using R? This is not explicitly stated in the Methods section but is stated in the caption to Table 1.

5. Line 155: EAC has not been mentioned in text before here, though it is explained in the Discussion (line 312), I would write it out in full here as well.

6. Have you corrected for different sample sizes among populations when calculating expected Heterozygosity?

7. It is not clear exactly how the geographic distances were obtained, using the Ruler tool, by hand in Google Earth, or calculating the edge length of a shapefile? Would this be easy to replicate if it has been measured by hand?

8. Can you display the Evanno delta-K plot from STRUCTURE, and the BIC plot from DAPC that you used to determine the number of appropriate population clusters (K=4). These plots could be added to the Supplementary Material.

9. Line 238: “Our unrooted phylogeny based on Maximum likelihood inferences supports the structure analyses (K = 4) distinguishing populations between CNSW, SNSW, VIC and TAS (Figure 5).” Actually, your unrooted tree shows three clusters and a grade. You may wish to use midpoint rooting to assist with interpretation here. It’s hard to see what is going on with this figure. Having said that, I do not think that its valid to apply this evolutionary model to a SNP data set. RAxML is definitely not designed with SNP data in mind, and you would be better off using software that deals with the assumptions created by this type of data set. If you can’t do this, I would suggest removing this part altogether. It doesn’t really add anything unique to your conclusions.

10. Line 238: You state that the “Eden population in southern NSW is a monophyletic clade within the broader NSW group, instead of its own lineage (Figure 5).” This does not make grammatical sense with regard to monophyly, and needs clarification.

11. Line 245: “These results would indicate possible hybridisation between CNSW, SNSW and TAS but not VIC (see K=4, Figure 3).” Firstly, this seems more like Discussion than results, and needs to move to that section. Hybridisation is not the only source of genetic similarity, so do check your assumptions here, and expand on all possibilities unless you can test amongst them. Having said that, hybridisation is not the right term to apply to interbreeding among populations.

12. Line 271: “Furthermore, the smaller-sized seadragons in VIC also appear to have a darker colour pattern compared to NSW and TAS seadragons (see Figure 1).” I don’t think you should refer to colour based on the single images displayed, and those that lack a colour standard. There are so many uncontrolled variables here.

13. Lines 278-280: Think about removing phylogenetic inference, as suggested above.

14. Lines 286-288: You may wish to also cite [28] here, as it also showed that the Portsea/Flinders animals were distinct from NSW and TAS.

15. Line 288-290: “The Bassian Isthmus [29] which has acted as a barrier to marine dispersal for 20-25 kya during the last glacial maximum may have prevented mixing between these populations, during which Victorian weedy seadragons diverged in isolation.” I would acknowledge that this has already been suggested, which you state in your introduction, but do not link to here.

16. Line 292: “been ample opportunity for hybridisation”. Do you mean ample opportunity for secondary contact?

You don’t usually here about populations hybridising, rather different species hybridising.

17. Line 297-298: “we suggest the Victorian P. taeniolatus to be a subpopulation, alternatively weedy seadragons along southeast coast of Australia are forming a species complex [5,48].” It has already been demonstrated that, mitochondrially, the VIC animals are not the closest relatives of the east coast populations, so your interpretations here are somewhat skewed by the fact your sampling gaps increase the differences you see. You really have very little evidence for suggesting a species complex, and I suggest you remove this speculation from here and line 316-320. You may say it’s possible, but you have no evidence for this at present. Also, it would be prudent to refer to the study which has sampled them across the distributional range and did not find evidence for this!

18. Line 303: you should remove discussion about colour differences as it was not tested here.

19. Line 309: issues with ‘hybridisation’ as discussed above. You just mean interbreeding among populations, which is the foundation of the biological species concept.

20. Lines 321 – 333: In the first sentence of this paragraph you state the difficulty in comparing the genetic diversity of the SNP dataset to other studies on syngnathids which have used microsatellite sequence data. As the number of alleles does have an effect on expected Heterozygosity values I would urge caution in directly comparing He values between different species and studies where the number of alleles may differ, particularly when the total number of alleles for this study is likely to be much higher given that thousands of SNP loci were used.

21. Line 358: Again, remove terminology ‘hybrids’

22. Line 367: make ‘scientist’ plural

23. References- many do not italicize species names. this needs to be fixed.

Reviewer #2: Comments to authors

This manuscript reports RAD sequencing of individuals of common seadragons (Phyllopteryx taeniolatus) in the eastern part of their range from New South Wales, Tasmania and Victoria. The authors use the data to investigate population structuring and genetic diversity. Morphological measurements are taken to investigate morphological differentiation. The study contributes important data for this charismatic species and uses the usual population genetic tools to find substantial population structure and some evidence for morphological differences. I overall find the paper interesting but suggest to tone down some of the conclusions to better fit the data.

Main points

Species complex

The authors suggest in the discussion that Victorian and New South Wales populations may be reproductively isolated to form a species complex. This conclusion is in my opinion not supported by the data. As I outline below, the morphological differences need further investigation. The genetic differentiation between those populations is large as per FST but that per se does not imply complete reproductive isolation. The Structure analyses at K=2 indicate some potential shared ancestry components. For such a far reaching claim, one would need additional analyses explicitly showing the absence of gene flow and, as the authors acknowledge, ideally more than one sampling site in Victoria and from the remaining distribution.

The conclusion that Victorian populations should be separately managed should therefore be carefully considered. I also wonder what the recommendation would be about Tasmanian and New South Wales populations. According to the given scenario. they were possibly historically connected when the Bassian Strait was closed but now they are separated. What would be the recommendation for their management given that their differentiation towards New South Wales is smaller?

Morphological analysis

I find the conclusion that the New South Wales and Victorian populations are morphologically distinct premature. The main difference is a difference in body size. A difference in body size could reflect temperature differences or differences in demographics or age structure of the population or even just sampling younger individuals in Victoria. These possibilities should be at least carefully discussed.

We can expect that all other metrics scale allometrically with body size, hence it is not surprising to find further differences between the two populations. The analyses should be performed with measurements that were scaled by the body size.

It is further not clear if the study separated males and females, which do show sexual dimorphism at least in the body depth. It would be great to test here if there are other differences between males and females.

The discussion compares Tasmanian to New South Wales individuals to support evolutionary scenarios but this data was not included in the main analysis. The Tasmanian individuals either need to be included in the main analysis or this discussion needs to be explicit that it is speculative.

Lastly, comparing only New South Wales and Victoria may miss variation that is being seen in the genetics. I would be interested to see a PCA of the morphological measurements of all individuals including Tasmania. It would be interesting to see if morphological structure follows genetic structure.

Data availability

Information on sequencing statistics and genotyping metrics for each sample need to be given.

Please give the vcf or genepop files and fasta files of loci should be deposited in Data Dryad or uploaded as Supplementary Material if file size allows.

Minor comments

L.1 Title: I personally also like “weedy” seadragon too but common seadragon should be used to support the attempt to standardize names https://www.fishnames.com.au/.

L.4 Rephrase to “Population genomics of the weedy seadragon” would be more appropriate given the title

L.23 The IUCN citation (?) seems misplaced here.

L.23 Unclear, range wide genetic population data is available in Wilson, Stiller & Rouse 2017.

L.28 replace [] with ()

L.29 It is unclear how IBD can support the existence of 4 distinct genetic clusters, because IBD measures the degree of increasing genetic differentiation with geographic distance and does not delineate clusters

L.54 Spell out scientific name Syngnathidae first and then refer with the common name in the second sentence

L.65 missing )

L.74 Please cite the version of the IUCN assessment because the status of common seadragons has changed throughout different assessments. The IUCN status (now reversed to Least Concern) should also be given.

L.84 There is controversy in the exact age of the Last Glacial Maximum but 20-25 KY seems to old. Lewis, S. E., Sloss, C. R., Murray-Wallace, C. V., Woodroffe, C. D., & Smithers, S. G. (2013). Post-glacial sea-level changes around the Australian margin: a review. Quaternary Science Reviews, 74, 115–138.

L.100 Reference to Supp Table 2 with collection needed

L.107 Please specify how many individuals were measured from each locality.

L.123 What barcodes were used? Were they single or dual barcodes?

L.140 Please provide some detail of how STACKS calls SNPs as this is an important factor determining the number and quality of the raw SNPs. How was the single SNP chosen, randomly or according to best depth of coverage?

L.142 What was the requirement for depth of coverage?

L.144 Abbreviations should be defined as observed heterozygosity (Ho) etc.

L.152 How many replicates to assess significance?

L.159 The sentence seems incomplete.

L.165 Please also include a simple PCA that does not require setting a value of K. DAPC aims to maximize the distance between K groups and can often appear like heavier structuring than in the data. A simple PCA allows to look at clustering with very little assumptions.

L.165 I may be wrong here but DAPC is PCA based which means that it cannot handle missing data. How were missing data points replaced? What K was chosen for DAPC?

L.174 What dataset was this done on? SNPs only or RAD loci, were invariant RAD loci included?

L.176 How was the substitution model chosen? I assume the analysis was done on the concatenation of all RAD loci without partitioning?

L.185 What was the sample size for VIC and NSW? Common seadragons are sexually dimorphic with deeper bodies in the females - was this accounted for in the analyses?

L.196 Please provide a table of sequencing and SNP statistics for each individual. The main text also needs statistics on average missing proportions

L.197 Unclear which groups this FST value is comparing.

L.206 Please add the population description as in Table 3 to Table 2.

L.223 Is allelic richness useful for binary SNP markers?

L.233 I think it would be more logical to place the Mantel test directly after the FST.

L.233 and following Please use past tense to describe results.

L.237 DAPC typo

L.241 The interpretation of a monophyletic Eden population and its relationship to the NSW populations depends on the rooting. This needs to be reformulated. It is also not clear what “instead of its own lineage” means. Eden is its own lineage.

L.245 Given the concerns regarding the proposed species complex, “gene flow” would be the better term to use over “hybridisation”.

L.260 Figure legend should mention that only branches with 100 bootstrap support are annotated.

L.265 We would expect allometric scaling of most of these measurements. Where the measurements scaled by the total TL?

L.291 The isthmus opened later, detailed reconstructions and dates are in Lambeck, K., & Chappell, J. (2001). Sea level change through the last glacial cycle. Science, 292(5517), 679–686.

L.292 Gene flow is more appropriate in this context.

L.293 The FST values are large but they do not indicate complete isolation.

L.303 Please give the source of this information.

L.313 distant?

L.313 is almost identical to L296 and following without providing new evidence

L.329 write out microsatellites

L.328 This sentence is hard to follow.

Figure 1 Sydney is all CAPS, other localities not. Would be helpful to add sample size to the map.

Suppl Table 3 Some measurements have more then 10 decimal places which cannot possibly be the accuracy of the ImageJ measurements

6. PLOS authors have the option to publish the peer review history of their article (what does this mean?). If published, this will include your full peer review and any attached files.

Reviewer #1: No

Reviewer #2: No

---

## [Author Response · Author response to Decision Letter 0]

4 Sep 2020

Response to Reviewers:

Manuscript: PONE-D-20-17523

Genomic and morphological evidence of distinct subpopulations in the endemic common (weedy) seadragon Phyllopteryx taeniolatus (Syngnathidae) along the east coast of Australia

Reviewer #1:

Summary

Klanten et al., present a population genetics study on the weedy seadragon (Phyllopteryx taeniolatus) using SNP sequence data generated from ddRADseq. This work builds on previous studies that used mitochondrial and microsatellite sequence data. In this study the authors have contributed further evidence for significant population structuring at a regional scale that has implications for the management of this species. They have also incorporated morphological measurements as further evidence of population isolation and interpret these results along with the genetic data with reference to the historical Bassian Isthmus as a barrier to dispersal. 

The SNP data generated in this study have been assembled and filtered appropriately to produce a high-quality dataset for which to analyse genetic diversity and patterns among the populations. The authors have used multiple different analyses to test for genetic population structure, and these have been carried out appropriately for the dataset. The phylogenetic analyses are not suitable for this data set however, and should be removed (comments below). There is also widespread unjustifiable use of the word hybridisation, which needs to be removed. Otherwise, this is a robust contribution to understanding these iconic fish.

Response: We thank Reviewer 1 for their constructive comments. We accept the reviewers point about the phylogenetic tree (Figure 5) and have opted to remove this from our manuscript. It is unnecessary as our conclusions are well supported by the remaining analyses. 

We also edited the word hybridisation using terms such as ‘interbreeding’ 

Note: We added additional citations (suggested by reviewers) so the citation # in our manuscript have changed. 

Specific comments:

1. Line 59: “…(taxonomically known as the common seadragon)” isn’t correct as common names are not controlled by the ICZN. Suggest change to: (also known as the common…).

Response: Edited as suggested by reviewer. We also mention (Title and Abstract) both “common or weedy”.

2. Line 62: Ruby seadragons are thought to have prehensile tails (Rouse et al. 2017 First live records of….Mar Bio Records).

Response: Thank you for pointing this out, we changed the sentence to make sure it’s only weedy seadragons that have no prehensile tails. 

“Phyllopteryx taeniolatus are weak swimmers and lack the prehensile tail of seahorses ...”

We also added Rouse et al. 2007 [22] as a reference. 

3. Lines 77-88: You state that mtDNA showed strong population structure for weedy seadragons, and lowered genetic diversity on the east coast. Yet at the end of this paragraph you state that leafy seadragon genetic diversity was moderate to low, a similar pattern to the weedy. This last statement is in contrast with [28] which actually reports some areas with quite high diversity, but showed low diversity for the east coast. You either need to elaborate further on this complexity, or just shorten the final sentence of the paragraph so it does not include a species comparison.

Response: We edited the sentence so that it does not include a comparison. 

“Another study [34] on the leafy seadragon (Phycodurus eques), using both mtDNA and microsatellites, also found strong population structure especially between western and southern Australian populations, with high to low genetic diversity.”

4. Line 149: Were pairwise Fst values also calculated in Arlequin or using R? This is not explicitly stated in the Methods section but is stated in the caption to Table 1.

Response: We added “… in Arlequin ..” to clarify sentence.

5. Line 155: EAC has not been mentioned in text before here, though it is explained in the Discussion (line 312), I would write it out in full here as well.

Response: Eastern Australian Current (EAC) now written in full.

6. Have you corrected for different sample sizes among populations when calculating expected Heterozygosity?

Response: We used Nei’s calculation of heterozygosity using the R package pegas. Nei’s equation is as follows: H ^=n/(n-1)(1-∑_(i=1)^k▒p_i^2 )

This calculation does not correct for different sample sizes. Without this correction values calculated from small samples sizes are likely to be biased upward. More to the point we don’t emphasise heterozygosity comparisons between populations and only consider the global (overall) values as they relate to other species. 

7. It is not clear exactly how the geographic distances were obtained, using the Ruler tool, by hand in Google Earth, or calculating the edge length of a shapefile? Would this be easy to replicate if it has been measured by hand?

Response: we measured by hand using Ruler tool ‘Path’ in Google Earth. 

“… using Google Earth following the Eastern Australian Current (EAC) along the east coast to Tasmania using the Ruler tool Path by hand.”

8. Can you display the Evanno delta-K plot from STRUCTURE, and the BIC plot from DAPC that you used to determine the number of appropriate population clusters (K=4). These plots could be added to the Supplementary Material.

Response: Plots have now been added to Supporting information (S1 Fig A and B).

9. Line 238: “Our unrooted phylogeny based on Maximum likelihood inferences supports the structure analyses (K= 4) distinguishing populations between CNSW, SNSW, VIC and TAS (Figure 5).” Actually, your unrooted tree shows three clusters and a grade. You may wish to use midpoint rooting to assist with interpretation here. It’s hard to see what is going on with this figure. Having said that, I do not think that its valid to apply this evolutionary model to a SNP data set. RAxML is definitely not designed with SNP data in mind, and you would be better off using software that deals with the assumptions created by this type of data set. If you can’t do this, I would suggest removing this part altogether. It doesn’t really add anything unique to your conclusions.

Response: We removed any text referring to phylogenetic tree and Figure 5 as suggested above from our manuscript.

10. Line 238: You state that the “Eden population in southern NSW is a monophyletic clade within the broader NSW group, instead of its own lineage (Figure 5).” This does not make grammatical sense with regard to monophyly, and needs clarification.

Response: This sentence has been removed, as we removed all text relating to phylogenetic tree and Figure 5.

11. Line 245: “These results would indicate possible hybridisation between CNSW, SNSW and TAS but not VIC (see K=4, Figure 3).” Firstly, this seems more like Discussion than results, and needs to move to that section. Hybridisation is not the only source of genetic similarity, so do check your assumptions here, and expand on all possibilities unless you can test amongst them. Having said that, hybridisation is not the right term to apply to interbreeding among populations.

Response: Removed this sentence from results section.

12. Line 271: “Furthermore, the smaller-sized seadragons in VIC also appear to have a darker colour pattern compared to NSW and TAS seadragons (see Figure 1).” I don’t think you should refer to colour based on the single images displayed, and those that lack a colour standard. There are so many uncontrolled variables here.

Response: We removed this sentence. Furthermore, the whole section on morphology has changed as per Reviewer 2.

13. Lines 278-280: Think about removing phylogenetic inference, as suggested above.

Response: Removed from manuscript.

14. Lines 286-288: You may wish to also cite [28] here, as it also showed that the Portsea/Flinders animals were distinct from NSW and TAS.

Response: We added citation Wilson et al. 2017 [30]. 

“Of note is that all individuals from two sites in Victoria (Flinders Bay and Portsea) are highly distinct, genetically and to a degree morphologically, to the main populations of central and southern NSW, as well as Tasmania [see also 30].”

15. Line 288-290: “The Bassian Isthmus [29] which has acted as a barrier to marine dispersal for 20-25 kya during the last glacial maximum may have prevented mixing between these populations, during which Victorian weedy seadragons diverged in isolation.” I would acknowledge that this has already been suggested, which you state in your introduction, but do not link to here.

Response: We added Wilson et al 2017 [30] in citation to this sentence.

“The Bassian Isthmus [30–32] which has acted as a barrier to marine dispersal …”

16. Line 292: “been ample opportunity for hybridisation”. Do you mean ample opportunity for secondary contact? You don’t usually here about populations hybridising, rather different species hybridising.

Response: changed and used gene flow

“Since the opening of the Bassian Isthmus, approximately 14-25 kya, there has been ample opportunity for gene flow, therefore the question arises …”

17. Line 297-298: “we suggest the Victorian P. taeniolatus to be a subpopulation, alternatively weedy seadragons along southeast coast of Australia are forming a species complex [5,48].” It has already been demonstrated that, mitochondrially, the VIC animals are not the closest relatives of the east coast populations, so your interpretations here are somewhat skewed by the fact your sampling gaps increase the differences you see. You really have very little evidence for suggesting a species complex, and I suggest you remove this speculation from here and line 316-320. You may say it’s possible, but you have no evidence for this at present. Also, it would be prudent to refer to the study which has sampled them across the distributional range and did not find evidence for this!

Response: We have changed this section and the next and hypothesis that by extending the study in future to include individuals from the entire distribution range and using genome-wide SNPs, this study has increased the power of analysis both by including a large number of markers, and by having markers in more biologically relevant regions of the genome. We would like to point out that mitochondrial data frequently has a different gene tree from nuclear data and is a tiny subset of the genes available for divergence. In addition, to having many more loci (both mt & nDNA), nuclear markers provide a better picture of divergence because they both code for more proteins, and are not only matrilinear, but are less affected by introgression and diverge faster.

We agree that this study alone does not provide sufficient evidence that they are different species, but our populations (and particularly Victoria) are quite divergent with FST values 

well into what separates species in many other organisms. This in fact is important for future management and conservation.

Please see page 14 in MS where we elaborate on this issue.

18. Line 303: you should remove discussion about colour differences as it was not tested here.

Response: We removed this section.

19. Line 309: issues with ‘hybridisation’ as discussed above. You just mean interbreeding among populations, which is the foundation of the biological species concept.

Response: Changed to ‘.. interbreeding .. ‘ 

20. Lines 321 – 333: In the first sentence of this paragraph you state the difficulty in comparing the genetic diversity of the SNP dataset to other studies on syngnathids which have used microsatellite sequence data. As the number of alleles does have an effect on expected Heterozygosity values I would urge caution in directly comparing He values between different species and studies where the number of alleles may differ, particularly

when the total number of alleles for this study is likely to be much higher given that thousands of SNP loci were used.

Response: We agree and have now added this statement to the end of that paragraph

“However, these data should be interpreted with caution. Theoretically, the maximum heterozygosity derived from biallelic loci (such as the SNPs employed here) is 0.5. On the other hand, heterozygosity at microsatellite loci can theoretically reach 1.0.”

21. Line 358: Again, remove terminology ‘hybrids’

Response: Removed and replaced with “interbreeding”

22. Line 367: make ‘scientist’ plural

Response: Corrected

23. References- many do not italicize species names. this needs to be fixed.

Response: Fixed all references, thank you for pointing out.

Reviewer #2

Main points

Species complex:

The authors suggest in the discussion that Victorian and New South Wales populations may be reproductively isolated to form a species complex. This conclusion is in my opinion not supported by the data. As I outline below, the morphological differences need further investigation. The genetic differentiation between those populations is large as per FST but that per se does not imply complete reproductive isolation. The Structure analyses at K=2 indicate some potential shared ancestry components. For such a far reaching claim, one would need additional analyses explicitly showing the absence of gene flow and, as the authors acknowledge, ideally more than one sampling site in Victoria and from the remaining distribution. The conclusion that Victorian populations should be separately managed should therefore be carefully considered. I also wonder what the recommendation would be about Tasmanian and New South Wales populations. According to the given scenario. they were possibly historically connected when the Bassian Strait was closed but now they are separated. What would be the recommendation for their management given that their differentiation towards New South Wales is smaller?

Response: We thank Reviewer 2 for their constructive comments. 

As Reviewer 1 (point 17 above) also pointed to this issue, we clarified that our data has limited evidence, for the moment, for a species complex, and we suggest that much more sampling (across its entire range) using genome-wide SNPs will be needed. We would like to leave this hypothesis in our manuscript as a possibility (assumption) that it may occur. It will open dialogue and direct to more studies into this species. We have elaborated on this extensively in our discussion see p.14 in MS (also see our extensive response to Reviewer 1 above).

Regarding management we are suggesting that each state should manage their populations in particular VIC.

Morphological analysis:

I find the conclusion that the New South Wales and Victorian populations are morphologically distinct premature. The main difference is a difference in body size. A difference in body size could reflect temperature differences or differences in demographics or age structure of the population or even just sampling younger individuals in Victoria. These possibilities should be at least carefully discussed. 

We can expect that all other metrics scale allometrically with body size, hence it is not surprising to find further differences between the two populations. The analyses should be performed with measurements that were scaled by the body size.

It is further not clear if the study separated males and females, which do show sexual dimorphism at least in the body depth. It would be great to test here if there are other differences between males and females.

The discussion compares Tasmanian to New South Wales individuals to support evolutionary scenarios but this data was not included in the main analysis. The Tasmanian individuals either need to be included in the main analysis or this discussion needs to be explicit that it is speculative.

Lastly, comparing only New South Wales and Victoria may miss variation that is being seen in the genetics. I would be interested to see a PCA of the morphological measurements of all individuals including Tasmania. It would be interesting to see if morphological structure follows genetic structure.

Response: We have now analysed morphological differences via examination of external metrics vs total length (biplots with linear trendlines, see S4 Fig) and via ordination LDA and PCA (Fig 5 A and B). We removed Table 3 and re-wrote the morphology sections in Methods and Results.

The biplots suggest that several metrics differ between Victorian individuals and both NSW and Tasmania, supporting our statement that some morphological differences were apparent, by removing ontogenetic confounding and suggesting overall shape differences. We now note this and that all measured individuals were adults. 

Unfortunately, not all specimens were sexed so we comment on this too. We were able to determine sex for over half the specimens (29 individuals out of total 45 measured), so could not fully analyse sex differences in morphology, but argue against that different sex ratios at each location are driving differences in shape. 

We included Tasmanian samples (n=5 only) separately in biplots and both LDA and PCA plots (see above). 

Data availability:

Information on sequencing statistics and genotyping metrics for each sample need to be given. Please give the vcf or genepop files and fasta files of loci should be deposited in Data Dryad or uploaded as Supplementary Material if file size allows.

Response: We have added # of reads and % missing loci for each individual to S1 Table. Also, we have now uploaded a vcf and genepop file as S1 Data and S2 Data as supporting data files. We added following sentence:

“Fastq sequence files were deposited in NCBI’s Sequence Read Archive (accession numbers: SRR12395874-12395945) and the associated metadata is available at GEOME [36] and is provided in S1 Table. Data generated as vcf file (S1 Data) and as genepop file (S2 Data) are available as supporting information.”

Minor comments:

L.1 Title: I personally also like “weedy” seadragon too but common seadragon should be used to support the attempt to standardize names https://www.fishnames.com.au/.

Response: We agree that common seadragon is proper, however these animals in particular are known widely as ‘weedy’ and is used generally in the literature (see ref’s 23-25, 28 and 52), as well as the IUCN Red List. 

We have added “common” to the title as well as at the start of the Abstract, and do say that the proper name is common, see also our response to Reviewer 1.

L.4 Rephrase to “Population genomics of the weedy seadragon” would be more appropriate given the title

Response: Changed as suggested.

L.23 The IUCN citation (?) seems misplaced here.

Response: Removed whole sentence.

L.23 Unclear, range wide genetic population data is available in Wilson, Stiller & Rouse 2017.

Response: Removed that sentence see above.

L.28 replace [] with ()

Response: Corrected

L.29 It is unclear how IBD can support the existence of 4 distinct genetic clusters, because IBD measures the degree of increasing genetic differentiation with geographic distance and does not delineate clusters

Response: We removed sentence. Instead related our reults of genetic clusters to DAPC and Bayesian clustering analysis.

L.54 Spell out scientific name Syngnathidae first and then refer with the common name in the second sentence

Response: Corrected

L.65 missing )

Response: Corrected

L.74 Please cite the version of the IUCN assessment because the status of common seadragons has changed throughout different assessments. The IUCN status (now reversed to Least Concern) should also be given.

Response: Added sentence: “The weedy seadragon’s status on the IUCN Red List was changed in 2017 from “Near Threatened” (NT) to “Least Concern” (LC) but their population trend is still considered as “Decreasing”

(https://dx.doi.org/10.2305/IUCN.UK.2017-2.RLTS.T17177A67624517.en).” 

L.84 There is controversy in the exact age of the Last Glacial Maximum but 20-25 KY seems to old. Lewis, S. E., Sloss, C. R., Murray-Wallace, C. V., Woodroffe, C. D., & Smithers, S. G. (2013). Post-glacial sea-level changes around the Australian margin: a review. Quaternary Science Reviews, 74, 115–138.

Response: Thank you for the REF which we added now. We would like to add that this particular paper concentrated on the post-glacial sea level fluctuations and does mention several times that the exact LGM times for Tasmania/Vic are controversial. 

According to Lambeck and Chappell 2001 as you also mentioned below (added this REF as well), the peak of LGM was around 25kya.

We based our dates on the isthmus (land bridge) that existed when VIC and TAS were connected (thus separating marine organisms between east and west coast) see also Lambeck and Chappell 2001. The opening of land bridge to the east was about 14 kya suggested by Lambeck and Chapell’s 2001 Fig 4. 

L.100 Reference to Supp Table 2 with collection needed

Response: Added. “In total 82 P. taeniolatus were sampled for genetic analysis between astral summer 2014/15 – 2018 (S1 Table).”

L.107 Please specify how many individuals were measured from each locality.

Response: Added sentence and provide full data now in S2 Table (morphology).

“In total we measured 45 individuals, 20 each from NSW and VIC and 5 from TAS (S2 Table).”

L.123 What barcodes were used? Were they single or dual barcodes?

Response: Sentence added. “The RADSeq data generated by AGRF used single barcodes that are incorporated into the names of the Fastq files for each individual.” 

L.140 Please provide some detail of how STACKS calls SNPs as this is an important factor determining the number and quality of the raw SNPs. How was the single SNP chosen, randomly or according to best depth of coverage?

Response: We have added a reference to the manuscript Rochette et al. (2019) that explains in section 2.8 the Bayesian genotype caller employed by STACKs V2 to call SNPs. We used the ‘write_single_snp’ option which outputs the first SNP per locus and the default setting which for min coverage with is 3x per individual which resulted in a mean coverage of 21.1 x per individual per locus. See text below. 

The manuscript now says:

“Demultiplexing of the sequence data and initial quality control was conducted at AGRF with STACKS v.1.41 [36,37] using default settings. Only sequences with exact barcodes were retained and sequences were trimmed to 140 bp. All other data processing took place at the Genomics and Bioinformatics Cluster at the University of Central Florida using STACKS v.2.0b. Loci were assembled using the denovo pipeline using default settings except we allowed for four mismatches between stacks within individuals (-M) and four mismatches between sample loci when building the catalog (-n). See Rochette et al. [39] for a detailed explanation of the Bayesian genotype caller employed in STACKS.

Stacks in which the numbers of reads were more than two standard deviations above the mean were assumed to be repetitive elements and removed from the data set. To eliminate poor quality samples, those individuals with fewer than 200,000 reads or less than 60% of loci were excluded from further analysis, resulting in a final sample size of 72 individuals. Using the ‘populations’ function in STACKS, we eliminated those loci with observed heterozygosity greater than 0.7 (--max-obs-het) and applied a minor allele frequency cut-off of 0.05 (--min-maf). We retained only those loci that amplified in ≥ 80% of individuals (-r). Output files were created by implementing the ‘write_single_snp’ option which outputs the first SNP per locus.”

And the results read:

“A total of 19,608 loci were resolved across 72 individuals of which 7462 were variable and passed filtering. Samples had an average coverage 21 reads per SNP with a data matrix that was 92.6% complete (S1 Table for % missing loci per individual).”

Please see comments below for more information by author of STACKs.

L.142 What was the requirement for depth of coverage?

Response: The default settings for STACKs is 3x. We used the default which results in an average coverage of 21x per individual per locus. This decision was made after a conversation with the authors of STACKs Julian Catchen.

From the Author of Stacks: “There is no minimum depth to call SNPs. This has always been handled by 

the SNP model, which calculates the likelihood of each position being a heterozygote and a homozygote (while incorporating a prior reflecting the allele frequencies in the wider populations). The model then weighs 

the evidence for a heterozygous versus a homozygous position using a likelihood ratio test resulting in a p-value. 

If you want to require more evidence, that is more reads, to support a particular SNP call, you can increase the required p-value to consider the likelihood ratio test statistically significant (--alpha flag to gstacks). The default p-value is (of course) 0.05, but you can lower this, which will require more evidence for a call to be declared 

statistically significant. 

Using -m at the population-program stage to randomly drop alleles that have already proven to be statistically significant is a bad idea and will result in random allele drop out. I regret ever making this option 

available and removed it when given the chance with the 2.0 release.”

https://groups.google.com/forum/#!topic/stacks-users/IzeSy3nw5Bg 

L.144 Abbreviations should be defined as observed heterozygosity (Ho) etc.

Response: Added

L.152 How many replicates to assess significance?

Response: We are unsure where this ‘replicates to assess significance’ is? 

Line 151-153 reads:

 “…. comparisons according to Benjamini and Yekutieli [37] by controlling false discovery rates using the p.adjust function in R. In addition, we tested for Isolation by Distance [38] using a Mantel test and 100,000 permutations, as implemented in the R package ade4 v.1.7-15 [39].”

As far as we can tell there are no missing # of replicates in this section. 

L.159 The sentence seems incomplete.

Response: Edited: “To explore the possibility of population structuring, we used two methods, Bayesian clustering algorithms of STRUCTURE and Discriminant Analysis of Principle Components (DAPC).”

L.165 Please also include a simple PCA that does not require setting a value of K. DAPC aims to maximize the distance between K groups and can often appear like heavier structuring than in the data. A simple PCA allows to look at clustering with very little assumptions.

Response: We have now included PCAs for our major groups in supplemental materials (S3 Fig). These do indeed show less clustering compared to the DAPC plots. However, multiple lines of evidence support the DAPC clusters including F-statistics and STRUCTURE analyses so the major conclusions of our study have not changed. 

L.165 I may be wrong here but DAPC is PCA based which means that it cannot handle missing data. How were missing data points replaced? What K was chosen for DAPC?

Response: Agreed. We’ve added the following text. 

“Because PCA analyses require no missing data (and DAPC uses PCA) we replaced missing data with the mean allele frequency for each locus using the method scaleGen in ADEGENET.”

A k of 4 was chosen for DAPC, see the S1 Fig for the BIC graph used to determine this. 

L.174 What dataset was this done on? SNPs only or RAD loci, were invariant RAD loci included?

Response: The phylogenetic tree has been removed from the paper as per Reviewer 1.

L.176 How was the substitution model chosen? I assume the analysis was done on the concatenation of all RAD loci without partitioning?

Response: The phylogenetic tree has been removed from the paper (see above).

L.185 What was the sample size for VIC and NSW? Common seadragons are sexually dimorphic with deeper bodies in the females - was this accounted for in the analyses?

Response: See also Reviewer 1 above, we changed the morphology section. We added sex where it was possible.

L.196 Please provide a table of sequencing and SNP statistics for each individual. The main text also needs statistics on average missing proportions

Response: These data have been added to S1 Table and a line in the text has been added, only 7.4% missing. “Samples had an average coverage of 21 reads per SNP with a data matrix that was 92.6% complete (S1 Table for % missing loci per individual).”

L.197 Unclear which groups this FST value is comparing. 

Response: Added to new S3 Table A (AMOVA results).

L.206 Please add the population description as in Table 3 to Table 2.

Response: Table 3 (morphology students t-test) has been removed and instead we have plotted all measurements against total length to eliminate allopatric patterns. 

L.223 Is allelic richness useful for binary SNP markers?

Response: We used this so other studies can compare. Some studies use Ho and/or He but a few terrestrial studies use allelic richness. During the course of writing the MS and comparing to other studies using SNPs (mainly terrestrial studies like cattle, horse etc) most authors used allelic richness more often than He or Ho. We attempted to show allelic richness of seadragons for future comparisons to other organisms. 

L.233 I think it would be more logical to place the Mantel test directly after the FST.

Response: Edited result section moved Mantel test after pairwise Fst.

L.233 and following Please use past tense to describe results.

Response: corrected

L.237 DAPC typo

Response: corrected

L.241 The interpretation of a monophyletic Eden population and its relationship to the NSW populations depends on the rooting. This needs to be reformulated. It is also not clear what “instead of its own lineage” means. Eden is its own lineage.

Response: Removed this sentence as we removed the phylogenetic tree as per Reviewer 1. 

L.245 Given the concerns regarding the proposed species complex, “gene flow” would be the better term to use over “hybridisation”.

Response: Removed this sentence as per Reviewer 1 and in discussion we edited to ‘interbreeding’.

L.260 Figure legend should mention that only branches with 100 bootstrap support are annotated.

Response: Figure 5 was removed as per Reviewer 1

L.265 We would expect allometric scaling of most of these measurements. Where the measurements scaled by the total TL?

Response: Added S4 Fig plotting all 15 measurements against total length. In discussion:

“While individuals from Victoria were somewhat smaller, biplots (S4 Fig) indicate that shaped differences summarised in Fig 5A were not simply length-related. Most NSW individuals were not sexed but it is unlikely skewed sex ratios of samples by location can account for shape differences.”

L.291 The isthmus opened later, detailed reconstructions and dates are in Lambeck, K., & Chappell, J. (2001). Sea level change through the last glacial cycle. Science, 292(5517), 679–686.

Response: Added citation Lambeck & Chappel clearly state that the land bridge was closed effectively until ca. 14kya but started to break up from 14kya albeit fluctuating.

L.292 Gene flow is more appropriate in this context.

Response: Corrected

L.293 The FST values are large but they do not indicate complete isolation.

Response: For SNP data there are indeed very large. See more of our explanation above for Reviewer 1. We have edited the sentence and also added Wilson et al 2017.

L.303 Please give the source of this information.

Response: we removed colour patterns as suggested by Reviewer 1.

L.313 distant?

Response: Corrected 

L.313 is almost identical to L296 and following without providing new evidence

Response: Edited, this whole section has been re written see p.14 in our MS. 

L.329 write out microsatellites

Response: Corrected

L.328 This sentence is hard to follow.

Response: Edited: “This low genetic diversity in both the endemic weedy (this study) and leafy [34] seadragons stands in comparison with other members of the family Sygnathidae, which have moderate to high heterozygosity. Specifically, He ranged from 0.452 – 0.949 in five studies of widespread seahorses (Hippocampus sp.; [56–59]) and from 0.752 – 0.958 in four widespread pipefishes (Syngnathus sp.; [60–63]).”

Figure 1 Sydney is all CAPS, other localities not. Would be helpful to add sample size to the map.

Response: Corrected, and sample sizes for each population have been added to legend of Fig 1.

Suppl Table 3 Some measurements have more then 10 decimal places which cannot possibly be the accuracy of the ImageJ measurements

Response: Corrected this table is now S2 Table.

---

## [Decision Letter · Decision Letter 1]

22 Sep 2020

PONE-D-20-17523R1

Genomic and morphological evidence of distinct populations in the endemic common (weedy) seadragon Phyllopteryx taeniolatus (Syngnathidae) along the east coast of Australia

PLOS ONE

Dear Dr. Klanten,

Thank you for submitting your manuscript to PLOS ONE. After careful consideration, we feel that it has merit but does not fully meet PLOS ONE’s publication criteria as it currently stands. Therefore, we invite you to submit a revised version of the manuscript that addresses the points raised during the review process.

We look forward to receiving your revised manuscript.

Kind regards,

Tzen-Yuh Chiang

Academic Editor

PLOS ONE

Reviewers' comments:

Reviewer's Responses to Questions

**Comments to the Author**

1. If the authors have adequately addressed your comments raised in a previous round of review and you feel that this manuscript is now acceptable for publication, you may indicate that here to bypass the “Comments to the Author” section, enter your conflict of interest statement in the “Confidential to Editor” section, and submit your "Accept" recommendation.

Reviewer #2: All comments have been addressed

2. Is the manuscript technically sound, and do the data support the conclusions?

Reviewer #2: Partly

3. Has the statistical analysis been performed appropriately and rigorously? 

Reviewer #2: Yes

4. Have the authors made all data underlying the findings in their manuscript fully available?

Reviewer #2: Yes

5. Is the manuscript presented in an intelligible fashion and written in standard English?

Reviewer #2: Yes

6. Review Comments to the Author

Reviewer #2: Klanten et al. have done a careful job at addressing the reviewers’ concerns and have added useful figures and additional material that improve the manuscript. My remaining concerns are regarding the still strong emphasis on a species complex, for which I do not see sufficient evidence. There are also questions regarding the PCA, which I think is a simple plotting error.

1) Both reviewer 1 and myself have raised concerns about the suggestion of a species complex in Victoria and New South Wales. The authors have made modifications in calling for extended sampling and the improved resolution that nuclear markers can give, which are useful additions. However, the text still strongly implies that there is a species complex but that separate species are “not yet advocated”. The presented evidence based on allele frequency differences analyzed in a number of ways is not strong enough to draw that conclusion. This needs justification from the literature. As I have pointed out in my last review, this view is omitting the results from Structure at K=2 that does show shared ancestry components between Victoria and the eastern sites. If there was reproductive isolation, one would expect completely separate ancestry proportions. The Structure results need to be mentioned.

2) Along the same vein, there are also multiple mentions of “very restricted to no gene flow” or similar. This lies on the assumption that FST values reflect migration rates. In practice, this relationship is not well established because there are also other factors which impact allele frequency differences (see for example McCauley DE, Whitlock MC. “Indirect measures of gene flow and migration: FST ≠ 1/(4Nm + 1)” Heredity. 1999;82:117–125). It would be more appropriate to refer to “significant allele frequency differences” or “significant genetic differentiation”, because this is what has actually been measured with FST.

3) I thank the authors for including the PCA analysis. I think there may have been a mistake in the PCA plotting. I wondered how such clear clustering in DAPC can result in the PCA plot as shown in Figure S3, where several individuals were not clustering with their parent population. If that was true this would be concerning because PCA and DAPC are conceptually very similar. I repeated the analysis (on genepop file with four population groups separated) and found a clear assignment of individuals to populations (see code and plot below). The structure of the PCA looks similar to Figure S3 but it looks like the coloring of the individuals according to their populations is off in Figure S3. There must be a glitch in the R script that the authors used. Good news is that the PCA will support the DAPC results much better and the authors could consider to include it in the main text (up to the authors).

(The screenshots will likely not appear in the online form, so please see the attached PDF version of the reviewer comments)

Minor comments

L.67 Not sure if reference 19 is appropriate for the number of Syngnathidae species; Reference 18 Eschmeyer actually lists 321 valid species

L.129ff. May be more readable as complete sentences?

L.219 Typo in “Principal components”

L.230 Typo in “patterns”

L.231 Population code “SYD” has not been defined before.

L.258 Unclear what the difference between FST = 0.561 and FST = 0.630 (L.262) is.

L.289 Move “(FST ranging 0.636– 0.703)” to the end of the sentence to include TAS

L.313 Reference to Supplementary Figure S1 needed.

L.370 Past tense? “were highly distinct”

L.373 Move references to after “as a barrier to marine dispersal for 20-25 kya

374 during the last glacial maximum (LGM)”

L.376 The strait opened 14 ka ago according to Lambeck & Chappell, so gene flow may have been reinstated between 14 kya and today, not 14-25 kya.

L.379 Type in “shape differences”

L.380 It does not seem obvious why sex differences would not affect the observed differences.

L.383 It is more appropriate to refer to allele frequency differences than gene flow (see above).

L.386 Need a reference that the found differentiation is extraordinarily high.

L.388 This indicates that some of the RAD loci obtained the mitochondrial genome. Is this likely? Intuitively, I would think that is unlikely to obtain mitochondrial RAD loci but maybe this is different in this case. If there were mitochondrial loci in the dataset, these would have to get filtered or modeled separately from nuclear loci because they are haploid and have a different effective population size.

L.398 This paragraph seems isolated and the point is not clear.

L.405 Unclear which populations “this division” refers to. The previous sentence states that TAS and NSW populations were historically connected, not divided. The following sentence also talks about similarities, rather than divisions.

L.415 change to “(see also [30])”?

L.435 Reference needed: “less resilient under normal circumstances, but instead have a smaller gene pool due to smaller population sizes and restricted distribution ranges”

L.436 It is more appropriate to refer to allele frequency differences than gene flow (see above).

L.440 Reference needed: “increase in anthropogenic stressors resulting in habitat loss (kelp and seagrass) and an increase of sea surface temperature”

L.458 It is more appropriate to refer to allele frequency differences than gene flow (see above).

7. PLOS authors have the option to publish the peer review history of their article (what does this mean?). If published, this will include your full peer review and any attached files.

Reviewer #2: No

---

## [Author Response · Author response to Decision Letter 1]

5 Nov 2020

Response to Reviewer Comments:

Klanten et al. have done a careful job at addressing the reviewers’ concerns and have added

useful figures and additional material that improve the manuscript. My remaining concerns are

regarding the still strong emphasis on a species complex, for which I do not see sufficient

evidence. There are also questions regarding the PCA, which I think is a simple plotting error.

1) Both reviewer 1 and myself have raised concerns about the suggestion of a species

complex in Victoria and New South Wales. The authors have made modifications in

calling for extended sampling and the improved resolution that nuclear markers can give,

which are useful additions. However, the text still strongly implies that there is a species

complex but that separate species are “not yet advocated”. The presented evidence

based on allele frequency differences analyzed in a number of ways is not strong

enough to draw that conclusion. This needs justification from the literature. As I have

pointed out in my last review, this view is omitting the results from Structure at K=2 that

does show shared ancestry components between Victoria and the eastern sites. If there

was reproductive isolation, one would expect completely separate ancestry proportions.

The Structure results need to be mentioned.

Response: Structure Harvester analyses indicate the most likely K is four. This is supported by DAPC and in part by AMOVA. We added the following text “Each population harboured private alleles including CNSW (1721), SNSW (348), TAS (563), and VIC (3918). While none of these were fixed a substantial proportion reached an allele frequency of 0.9 meaning they were nearly fixed (CNSW: 1200, 70%; SNSW: 128, 37%; TAS: 161, 29%; VIC: 1385, 35%).” The pattern presented in the Structure plots when looking at all values of K can be explained by incomplete lineage sorting within recently diverged lineages. Further, we would not expect completely separate ancestry proportions, as it takes time for such differences to fix, and these were likely in full contact during the last interglacial period (about 120 kya), and intermittently until the glacial maximum about 40 kya. Finally, due to our results and west to east current of the Bass Strait, there may be occasional unidirectional introgression of Victorian genes to the eastern populations, but the reverse is highly unlikely, which is why we specifically noted this population even though the others also demonstrate high differentiation.

We agree with the Reviewer that more evidence is still needed to confirm distinct species but note that the genetic data, backed up by morphological, are entirely consistent, with the hypothesis that these are diverging populations that are not randomly mating and possibly not mating at all. These, we believe are important considerations for conservation. 

 We feel that it would be remiss to ignore a reasonable scientific discussion of our results and their interpretation, which we strongly feel should include a hypothesis of a subspecies and how we might further test this. Apart from the evolutionary and ecological implications, conservation management of this iconic species will depend on species population structure across its range. 

Therefore, our Discussion includes such speculation and a way forward. However, note that we do not use these terms (subspecies) in our Title, Abstract, and Results.

We would also like to draw the Reviewers attention to a few recent articles that support genome level studies to delineate species-level structure (subspecies, distinct lineages etc) examples are: 

1. Baumsteiger et al. 2017, Genomics clarifies taxonomic boundaries in a difficult species complex. PLoS ONE 12(12): e0189417. https://doi.org/10.1371/journal.pone.0189417. 

2. Roux et al. 2016, Shedding Light on the Grey Zone of Speciation along a Continuum of Genomic Divergence. PLoS Biol 14(12): e2000234. doi:10.1371/journal.pbio.2000234.

3. Beheregaray et al. 2017, Genome-wide data delimits multiple climate-determined species ranges

in a widespread Australian fish, the golden perch (Macquaria ambigua) Mol. Phylog. Evol 111, 65-75.

2) Along the same vein, there are also multiple mentions of “very restricted to no gene flow”

or similar. This lies on the assumption that FST values reflect migration rates. In

practice, this relationship is not well established because there are also other factors

which impact allele frequency differences (see for example McCauley DE, Whitlock MC.

“Indirect measures of gene flow and migration: FST ≠ 1/(4Nm + 1)” Heredity .

1999;82:117–125). It would be more appropriate to refer to “significant allele frequency

differences” or “significant genetic differentiation”, because this is what has actually been

measured with FST.

Response: McCauley and Whitlock’s argument is that one should not directly measure gene flow using Wright’s F-statistics, and we do not do so in this paper. We accept that such a calculation would be inaccurate, however, references to the direction and loose magnitude of link between gene flow and genetic differentiation are still valid. We have altered much of the wording to reduce confusion but have retained it where appropriate. 

3) I thank the authors for including the PCA analysis. I think there may have been a mistake

in the PCA plotting. I wondered how such clear clustering in DAPC can result in the PCA

plot as shown in Figure S3, where several individuals were not clustering with their

parent population. If that was true this would be concerning because PCA and DAPC are

conceptually very similar. I repeated the analysis (on genepop file with four population

groups separated) and found a clear assignment of individuals to populations (see code

and plot below). The structure of the PCA looks similar to Figure S3 but it looks like the

coloring of the individuals according to their populations is off in Figure S3. There must

be a glitch in the R script that the authors used. Good news is that the PCA will support

the DAPC results much better and the authors could consider to include it in the main

text (up to the authors).

(The screenshots will likely not appear in the online form, so please see the attached

PDF version of the reviewer comments)

Response: We thank Reviewer 2 for catching this! We corrected our code (apparently an older table was used where some of the samples were incorrectly coded). 

Minor comments

L.67 Not sure if reference 19 is appropriate for the number of Syngnathidae species;

Reference 18 Eschmeyer actually lists 321 valid species

Response: Corrected and removed REF 19

L.129ff. May be more readable as complete sentences?

Response: Edited

L.219 Typo in “Principal components”

Response: Corrected 

L.230 Typo in “patterns”

Response: Corrected 

L.231 Population code “SYD” has not been defined before.

Response: Corrected to NSW (typo as many samples measured were from Sydney area)

L.258 Unclear what the difference between FST = 0.561 and FST = 0.630 (L.262) is.

Response: We clarified the difference between the two Fst values. One was for all 9 pops (overall), the second Fst was for grouping the 9 pop’s into 4 areas.

L.289 Move “(FST ranging 0.636– 0.703)” to the end of the sentence to include TAS

Respose: Moved to the end of sentence

L.313 Reference to Supplementary Figure S1 needed.

Response: Added

L.370 Past tense? “were highly distinct”

Response: Corrected

L.373 Move references to after “as a barrier to marine dispersal for 20-25 kya 374 during the last glacial maximum (LGM)”

Response: Moved reference

L.376 The strait opened 14 ka ago according to Lambeck & Chappell, so gene flow may

have been reinstated between 14 kya and today, not 14-25 kya.

Response: Corrected

L.379 Type in “shape differences”

Response: Corrected

L.380 It does not seem obvious why sex differences would not affect the observed

differences.

Response: We clarified this by adding a sentence in the Results (most NSW samples were not sexed, and we had only 4 females in VIC see also S2 Table) and explained why we cannot report a skewed sex ratio in the discussion (we had only 10 females out of 45 specimens). They did not influence our linear trendline for body depth, however to resolve this we would need to measure more females especially in NSW where females can be easily distinguished in situ by body depth.

L.383 It is more appropriate to refer to allele frequency differences than gene flow (see

above).

Response: We thank the reviewer but can’t find a reference to gene flow on line 383. 

L.386 Need a reference that the found differentiation is extraordinarily high.

Response: Added Ref 54.

L.388 This indicates that some of the RAD loci obtained the mitochondrial genome. Is

this likely? Intuitively, I would think that is unlikely to obtain mitochondrial RAD loci but maybe

this is different in this case. If there were mitochondrial loci in the dataset, these would have to

get filtered or modeled separately from nuclear loci because they are haploid and have a

different effective population size.

Response: We have removed the references to this. We do believe some mitochondrial loci were sequenced, but haploid loci would have been filtered out when we only retained polymorphic sites.

L.398 This paragraph seems isolated and the point is not clear.

Response: We moved this sentence up, to where we talk about isolation distinguishing populations

L.405 Unclear which populations “this division” refers to. The previous sentence states

that TAS and NSW populations were historically connected, not divided. The following sentence

also talks about similarities, rather than divisions.

Response: We edited and moved this sentence up where we talk about Vic pop managed separately to NSW, TAS pop’s.

L.415 change to “(see also [30])”?

Response: Added now it’s REF 29

L.435 Reference needed: “less resilient under normal circumstances, but instead have

a smaller gene pool due to smaller population sizes and restricted distribution ranges”

Response: Added references [9, 13]

L.436 It is more appropriate to refer to allele frequency differences than gene flow (see

above).

Response: Edited to genetic differentiation 

L.440 Reference needed: “increase in anthropogenic stressors resulting in

habitat loss (kelp and seagrass) and an increase of sea surface temperature”

Response: Added references

L.458 It is more appropriate to refer to allele frequency differences than gene flow (see

above).

Response: We would like to keep ‘restricted gene flow’ as this section is in general terms and appropriate as it concerns management perspectives.

---

## [Decision Letter · Decision Letter 2]

23 Nov 2020

Genomic and morphological evidence of distinct populations in the endemic common (weedy) seadragon Phyllopteryx taeniolatus (Syngnathidae) along the east coast of Australia

PONE-D-20-17523R2

Dear Dr. Klanten,

We’re pleased to inform you that your manuscript has been judged scientifically suitable for publication and will be formally accepted for publication once it meets all outstanding technical requirements.

Kind regards,

Tzen-Yuh Chiang

Academic Editor

PLOS ONE

Additional Editor Comments (optional):

Reviewers' comments:

Reviewer's Responses to Questions

**Comments to the Author**

1. If the authors have adequately addressed your comments raised in a previous round of review and you feel that this manuscript is now acceptable for publication, you may indicate that here to bypass the “Comments to the Author” section, enter your conflict of interest statement in the “Confidential to Editor” section, and submit your "Accept" recommendation.

Reviewer #2: All comments have been addressed

2. Is the manuscript technically sound, and do the data support the conclusions?

Reviewer #2: Yes

3. Has the statistical analysis been performed appropriately and rigorously? 

Reviewer #2: Yes

4. Have the authors made all data underlying the findings in their manuscript fully available?

Reviewer #2: Yes

5. Is the manuscript presented in an intelligible fashion and written in standard English?

Reviewer #2: Yes

6. Review Comments to the Author

Reviewer #2: Thanks to the authors for addressing all comments and suggestions. I have no further suggestions for this manuscript.

7. PLOS authors have the option to publish the peer review history of their article (what does this mean?). If published, this will include your full peer review and any attached files.

Reviewer #2: No

---

## [Editor Report · Acceptance letter]

2 Dec 2020

PONE-D-20-17523R2 

Genomic and morphological evidence of distinct populations in the endemic common (weedy) seadragon *Phyllopteryx taeniolatus* (Syngnathidae) along the east coast of Australia 

Dear Dr. Klanten:

I'm pleased to inform you that your manuscript has been deemed suitable for publication in PLOS ONE. Congratulations! Your manuscript is now with our production department. 

Kind regards, 

on behalf of

Dr. Tzen-Yuh Chiang 

Academic Editor

PLOS ONE